# Strategyproof Reinforcement Learning from Human Feedback

**Thomas Kleine Buening**
ETH Zurich
ETH AI Center

**Jiarui Gan**
Department of Computer Science
University of Oxford

**Debmalya Mandal**
Department of Computer Science
University of Warwick

**Marta Kwiatkowska**
Department of Computer Science
University of Oxford

## Abstract

We study Reinforcement Learning from Human Feedback (RLHF) in settings where multiple labelers may strategically misreport feedback to steer the learned policy toward their own preferences. We show that existing RLHF algorithms, including recent pluralistic methods, are not strategyproof, and that even a single strategic labeler can cause arbitrarily large misalignment with social welfare. Moreover, we prove that, in the worst case, any strategyproof RLHF algorithm must perform $k$-times worse than the optimal policy, where $k$ is the number of labelers. This suggests a fundamental trade-off between incentive alignment (ensuring labelers report truthfully) and policy alignment (maximizing social welfare). To address this, we propose the Pessimistic Median of MLEs algorithm, which, under appropriate policy coverage assumptions, is approximately strategyproof and converges to the optimal policy as the number of labelers and samples increases. Our results apply to both contextual bandits and Markov decision processes.

## 1 Introduction

Reinforcement Learning from Human Feedback (RLHF) has become a widely used approach for aligning AI systems with human preferences. By leveraging human-labeled comparisons, RLHF enables policy optimization in applications such as robotics, recommendation systems, and Large Language Models (LLMs) [5, 21]. This approach has led to significant improvements in usability and alignment with intended objectives. However, RLHF also introduces new challenges, particularly in situations where preferences are diverse, subjective, and potentially in conflict.

Recently, pluralistic alignment—the challenge of aligning with the preferences of diverse individuals or groups—has emerged as an active area of research [33, 14, 22, 3, 6]. Unlike traditional reinforcement learning, which optimizes a policy according to a single, well-defined reward function, pluralistic settings require reconciling multiple perspectives. This raises questions about whose preferences should shape AI decisions and how to aggregate diverse inputs fairly and effectively [20, 11]. In such pluralistic settings, existing methods often optimize policies based on aggregated human preferences, implicitly assuming that labelers provide feedback truthfully. However, this perspective crucially neglects the incentives of labelers when their feedback directly impacts policy outcomes.

When human preferences influence the final policy, labelers (or groups of labelers) may have incentives to manipulate their feedback in ways that benefits them at the expense of broader alignment [30, 19]. For example, in the context of LLM fine-tuning, human labelers may systematically misreport preferences to amplify specific biases and reinforce narratives favorable to their views.[1]

---

[1] Naturally, this challenge is not exclusive to human labelers but can also arise, or even be amplified, when learning from synthetic feedback sources designed to influence model behavior.

39th Conference on Neural Information Processing Systems (NeurIPS 2025).

As a result, such strategic behavior threatens to distort the alignment process through self-serving feedback and can undermine the fairness, robustness, and effectiveness of the system [13, 2]. Despite its importance, this issue remains largely unaddressed in existing RLHF methodologies.

This paper aims to bridge this gap by studying RLHF through the lens of mechanism design and proposing solutions that ensure robustness against strategic feedback. We formalize the problem, analyze the conditions and the cost under which strategyproofness can be achieved, and propose an RLHF method to mitigate incentive misalignment while maintaining policy performance.

In summary, our main contributions are:

- We formally introduce the problem of offline RLHF with strategic human labelers, where each labeler potentially misreports preference labels to steer the final policy toward the maximization of their personal objectives, i.e., their reward function (Section 3). We focus on linear reward functions and social welfare maximization and study the tensions that arise between individual incentive alignment and policy alignment with social welfare.

- We show that existing RLHF methods are not strategyproof (Proposition 3.3), and even a single strategic labeler can almost arbitrarily degrade policy performance of existing methods (Proposition 3.4). Moreover, without additional assumptions, we find that any strategyproof RLHF method suffers from constant suboptimality (Theorem 3.5) and performs at least $k$-times worse compared to the optimal policy (Corollary 3.6), where $k$ is the number of distinct labelers. This points towards a fundamental trade-off between incentive alignment and policy alignment.

- We propose an RLHF method called Pessimistic Median of MLEs, which combines pessimistic estimates with a median rule to incentivize truthful preference reporting (Section 4). Interestingly, we find that Pessimistic Median of MLEs is *approximately* strategyproof due to the uncertainty in reward estimation. Notably, the incentive strength depends on the *uniform* policy coverage of each labeler's data. This stands in contrast to standard RLHF guarantees, which rely only on the coverage of the optimal policy. More precisely, under additional domain restrictions, we show:

  a) Pessimistic Median of MLEs is $\tilde{\mathcal{O}}(\kappa_i \sqrt{d/n})$-strategyproof for labeler $i$ where $\kappa_i$ quantifies the uniform policy coverage of labeler $i$ (Theorem 4.1).

  b) The computed policy's suboptimality is bounded by $\tilde{\mathcal{O}}(\sqrt{d/k} + \max_{i\in[k]} \kappa_i^* \cdot k \sqrt{d/n})$ where $\kappa_i^*$ is the optimal policy coverage of labeler $i$ (Theorem 4.2, Proposition 4.3, Corollary 4.5).

  We establish these results for both contextual bandits and Markov decision processes (Section 5).

## 2 Related Work

**Reinforcement Learning from Human Feedback.** RLHF has emerged as a powerful framework for aligning AI systems with human values by leveraging human feedback to guide policy learning [10, 5, 21]. Most relevant to our work is the growing literature on RLHF in settings with diverse and possibly conflicting preferences among individuals or demographic groups [7, 28, 37, 8]. Some of these works focus on maximizing the worst-case utility across labelers (or groups) [7, 28], whereas others optimize welfare functions such as the additive social welfare [37].

Some other recent work has also explicitly taken a social choice perspective on pluralistic alignment and studies how to ensure that methods for preference aggregation satisfy desirable properties inspired by social choice theory [11, 15, 1]. Importantly, these works, while related, assume truthful feedback and do not account for the incentives created by AI alignment. However, aggregating and trading-off preferences naturally invites strategic or malicious behavior, as labelers may manipulate the alignment process to more closely align the final policy with their own beliefs and goals. For example, Siththaranjan et al. [32] highlighted how standard RLHF methods implicitly aggregate preferences using the Borda count voting rule, which can create incentives for annotators to misreport their preferences to influence model behavior.

Another body of work considers robustness against adversarial corruptions in RLHF. Mandal et al. [24] assume that an $\varepsilon$-fraction of samples is adversarially manipulated, allowing for both manipulation of trajectory features and preference labels. Similarly, Bukharin et al. [4] and Cheng et al. [9] also consider the case where a fraction of samples is manipulated but restrict their attention to adversaries flipping preferences. This line of work differs from ours notably in both perspective (strategic vs. adversarial) and techniques (mechanism design vs. robust offline RL).

**Mechanism Design for RLHF.** Recently, several works have incorporated mechanism design principles into RLHF to incentivize truthful feedback [27, 34, 35]. These approaches design payment rules to align labelers' incentives, often extending VCG-style mechanisms to the RLHF problem. In contrast, we propose a strategy-robust RLHF method that does not rely on payments or other financial incentives, which are often impractical in real-world applications. Also closely related is the work of Hao and Duan [18], which studies an online RLHF framework where labelers sequentially provide preference feedback, aggregated using a linearly weighted average. Their approach focuses on identifying the most accurate labeler over time and adjusting the weights to incentivize truthful reporting. In contrast, we study the offline RLHF setting and do not impose a linear weighting assumptions on labelers. Moreover, unlike [18], we assume that labelers seek to influence the final policy rather than just an estimate of the aggregated preferences, which might better reflect real-world strategic behavior, where individuals care about the actual policy outcomes rather than intermediate preference estimates. In particular, as we will see, a more closely aligned reward function does not imply a more aligned policy.

## 3 Problem Formulation

We consider episodic Markov Decision Processes (MDPs) and the special case of contextual bandits. Let $\mathcal{M} = (\mathcal{S}, \mathcal{A}, \mathcal{P}, H, \rho)$ be an MDP without reward function, where $\mathcal{S}$ is the state space, $\mathcal{A}$ is the action space, $H$ is the horizon and $\rho$ is the initial state distribution. $\mathcal{P} = (\mathcal{P}_1, \ldots, \mathcal{P}_H)$ denotes the tuple of transition functions, where $\mathcal{P}_h : \mathcal{S} \times \mathcal{A} \to \Delta(\mathcal{S})$ determines the transitions in step $h \in [H]$. A history-independent policy $\pi = (\pi_h)_{h \leq H}$ maps from states to a distribution over actions in every time step, i.e., $\pi_h : \mathcal{S} \to \Delta(\mathcal{A})$, and we let $\Pi$ denote its policy space. A trajectory in MDP $\mathcal{M}$ is given by a sequence of actions and states $\tau = (a_1, s_2, a_2, \ldots, s_H, a_H)$. The MDP reduces to a contextual bandit problem when $H = 1$, in which case a trajectory consists only of the action taken in the initial state and the initial states are interpreted as contexts sampled from $\rho$.

**Multiple Labelers with Diverse Preferences.** We consider the situation where $k \geq 1$ many labelers provide preference data to the RLHF algorithm. In particular, each labeler $i \in [k]$ is associated with a reward function $r_i : \mathcal{S} \times \mathcal{A} \to \mathbb{R}$. The expected return of a policy $\pi$ w.r.t. a reward function $r$ is given by $V_r^\pi(s) := \mathbb{E}\big[\sum_{h=1}^{H} r(s_h, \pi_h(s_h)) \mid s_1 = s\big]$.[2] Accordingly, we define the *utility* of labeler $i \in [k]$ w.r.t. the initial state distribution $\rho$ and a policy $\pi$ as

$$J_i(\rho, \pi) := \mathbb{E}_{s \sim \rho}\left[V_{r_i}^\pi(s)\right].$$

Note that this simplifies to $J_i(\rho, \pi) = \mathbb{E}_{s \sim \rho}[r_i(s, \pi(s))]$ in the contextual bandit.

We focus on the linearly realizable case, where the reward function of each labeler is a linear function $r_\theta(s, a) = \langle \theta, \phi(s, a) \rangle$ of a known feature embedding $\phi$ of the state (i.e., context) and action.

**Assumption 1** (Linear Realizability). Every labeler's reward function $r_i$ is given by a linear function $r_{\theta_i^*}(s, a) := \langle \theta_i^*, \phi(s, a) \rangle$. Here, the reward parameter $\theta_i^*$ is sampled from $\{\theta \in \mathbb{R}^d : \|\theta\|_2 \leq B\}$ with $B > 0$ and $\phi$ is a known mapping with $\|\phi(s, a)\|_2 \leq L$ for all $(s, a) \in S \times A$.

**Offline RLHF.** We focus on the offline RLHF setting, where each labeler $i \in [k]$ is given a pre-determined set of $n$ examples $(s^{i,j}, \tau_0^{i,j}, \tau_1^{i,j})$ indexed by $j \in [n]$ where $s^{i,j}$ denotes the initial state and $(\tau_0^{i,j}, \tau_1^{i,j})$ are two subsequent trajectories. For each such example, labeler $i$ provides a preference label $o^{i,j} \in \{0, 1\}$, where the label $o^{i,j} = 0$ means that trajectory $\tau_0^{i,j}$ is preferred over $\tau_1^{i,j}$ given initial state $s^{i,j}$, and vice versa.[3]

We employ the widely used Bradley-Terry (BT) model under which a labeler with a reward parameter $\theta$ (i.e., reward function $r_\theta$) prefers trajectory $\tau_0 = (a_1, s_2, a_2, \ldots, s_H, a_H)$ over trajectory $\tau_1 = (\tilde{a}_1, \tilde{s}_2, \tilde{a}_2, \ldots, \tilde{s}_H, \tilde{a}_H)$ with probability

$$\mathbb{P}_\theta(o = 0 \mid s, \tau_0, \tau_1) := \frac{\exp(r_\theta(s, \tau_0))}{\exp(r_\theta(s, \tau_0)) + \exp(r_\theta(s, \tau_1))}, \tag{1}$$

where $r_\theta(s, \tau_0) := \sum_{h=1}^{H} r_\theta(s_h, a_h)$ is the total reward of the trajectory $\tau_0$ given initial state $s_1 = s$. In a contextual bandit, where each trajectory consists of a single action only, i.e., $\tau_0 = a_0$ and $\tau_1 = a_1$, the comparison model conveniently simplifies to $\mathbb{P}_\theta(o = 0 \mid s, a_0, a_1) \propto \exp(r_\theta(s, a_0))$.

---

[2] With slight abuse of notation we let $r(s, \pi(s)) = \mathbb{E}_{a \sim \pi(\cdot|s)}[r(s, a)]$ if the policy $\pi$ is stochastic.

[3] To ease notation, we assume that every labeler provides preferences for the same number of examples $n$. This can be straightforwardly relaxed at the cost of additional notation and slightly more cumbersome statements.

**Strategic Preference Labeling.** We assume that *if* labeler $i \in [k]$ provides their preferences *truthfully*, then the preference labels $o^{i,j}$ are sampled with respect to their true reward function $r_{\theta_i^*}$. Thus, the collected preference dataset under truthful labeling is given by

$$\mathcal{D}_i^* = (s^{i,j}, \tau_0^{i,j}, \tau_1^{i,j}, o^{i,j})_{j \leq n} \text{ with } o^{i,j} \sim \mathbb{P}_{\theta_i^*}(\cdot \mid s^{i,j}, \tau_0^{i,j}, \tau_1^{i,j}).$$

Since each labeler aims to more closely align the final policy with their personal preferences, a labeler may strategically manipulate labels to maximize their utility $J_i$. In this case, we assume that labeler $i$ samples preference labels according to a manipulated reward function $r_{\tilde{\theta}_i}$:

$$\tilde{\mathcal{D}}_i = (s^{i,j}, \tau_0^{i,j}, \tau_1^{i,j}, \tilde{o}^{i,j})_{j \leq n} \text{ with } \tilde{o}^{i,j} \sim \mathbb{P}_{\tilde{\theta}_i}(\cdot \mid s^{i,j}, \tau_0^{i,j}, \tau_1^{i,j}).$$

Note that the examples $(s^{i,j}, \tau_0^{i,j}, \tau_1^{i,j})$ remain fixed and only the preference labels change.[4]

Given a reported preference dataset $\mathcal{D} = (\mathcal{D}_1, \ldots, \mathcal{D}_k)$, an RLHF algorithm computes a policy $\hat{\pi}_{\mathrm{RLHF}}(\mathcal{D}) \in \Pi$. We omit the argument $\mathcal{D}$ when the dataset is clear from context. We want to highlight that it is unknown to the RLHF algorithm (and impossible to tell) whether the preference labels in the dataset were truthfully reported or not.

We can now define what it means for an RLHF algorithm to be robust against strategic manipulation (in the incentive alignment sense). In short, an RLHF method is *strategyproof* if truthfulness is an optimal strategy for every labeler irrespective of what the other labelers report.

**Definition 3.1** (Strategyproofness)**.** We say that the mapping $\hat{\pi}_{\mathrm{RLHF}}(\mathcal{D})$ is strategyproof if for all $i \in [k]$, other labelers' data $\mathcal{D}_{-i} = (\mathcal{D}_1, \ldots, \mathcal{D}_{-i}, \mathcal{D}_{i+1}, \ldots, \mathcal{D}_k)$ and deviation $\tilde{\theta}_i \neq \theta_i^*$ it holds that

$$\mathbb{E}_{o^{i,j} \sim \mathbb{P}_{\theta_i^*}} \left[ J_i\big(\hat{\pi}_{\mathrm{RLHF}}(\mathcal{D}_i^*, \mathcal{D}_{-i})\big) \right] \geq \mathbb{E}_{\tilde{o}^{i,j} \sim \mathbb{P}_{\tilde{\theta}_i}} \left[ J_i\big(\hat{\pi}_{\mathrm{RLHF}}(\tilde{\mathcal{D}}_i, \mathcal{D}_{-i})\big) \right].$$

This property is also commonly referred to as *dominant strategy incentive compatibility*.

We can relax the strict incentive constraint by allowing labelers to have a limited incentive to misreport, which provides us with the notion of $\varepsilon$-strategyproofness.

**Definition 3.2** ($\varepsilon$-Strategyproofness)**.** We say that the mapping $\pi_{\mathrm{RLHF}}(\mathcal{D})$ is $\varepsilon$-strategyproof with $\varepsilon > 0$ if for all $i \in [k]$, other labelers' data $\mathcal{D}_{-i}$ and deviation $\tilde{\theta}_i \neq \theta_i^*$ it holds that

$$\mathbb{E}_{o^{i,j} \sim \mathbb{P}_{\theta_i^*}} \left[ J_i\big(\pi_{\mathrm{RLHF}}(\mathcal{D}_i^*, \mathcal{D}_{-i})\big) \right] \geq \mathbb{E}_{\tilde{o}^{i,j} \sim \mathbb{P}_{\tilde{\theta}_i}} \left[ J_i\big(\pi_{\mathrm{RLHF}}(\tilde{\mathcal{D}}_i, \mathcal{D}_{-i})\big) \right] - \varepsilon.$$

A few comments are in place. The careful reader might wonder why we define strategyproofness at the *distributional level (ex ante)* rather than at the level of realized preference labels, e.g., by allowing labelers to flip preferences *after* sampling. The reason is that defining misreporting at the level of preference realizations instead of preference distributions would blur the line between strategic manipulation and post hoc noise correction, which is not the focus of our analysis. One can imagine that even a (conceptually) perfectly strategyproof algorithm would incentivize the labelers to flip preference realizations post hoc in an attempt to better teach the algorithm their reward function by correcting noise. Defining the labelers' strategies over preference distributions instead of realizations ensures a more meaningful comparison between truthful and strategic labeling and avoids these complications.

**Learning Objective.** We assume here that the set of labelers is representative of the population whose preferences we wish to to align to. Our objective is then to compute a policy maximizing the *average social welfare* given by

$$\mathcal{W}(\rho, \pi) := \frac{1}{k} \sum_{i=1}^{k} J_i(\rho, \pi).$$

In the following, we omit $\rho$ whenever the initial state distribution is clear from the context. Let $\pi^* := \mathrm{argmax}_{\pi \in \Pi} \mathcal{W}(\rho, \pi)$ be the optimal policy maximizing social welfare. The *suboptimality* of a policy $\pi$ is defined as

$$\mathrm{SubOpt}(\rho, \pi) := \mathcal{W}(\rho, \pi^*) - \mathcal{W}(\rho, \pi).$$

---

[4]While we choose to present our results for mislabeling within the class of BT models for ease of presentation, we want to highlight that our results on strategyproofness in Section 4.1 extend beyond the BT model and apply also when labelers misreport according to arbitrary preference distributions.

In addition to this standard notion of suboptimality, it can also be insightful to consider the multiplicative *approximation ratio* of a policy $\pi$ that is frequently studied in the computational social choice literature and given by the ratio

$$\alpha(\rho, \pi) \coloneqq \frac{\mathcal{W}(\rho, \pi)}{\mathcal{W}(\rho, \pi^*)}.$$

By definition, this ratio satisfies $\alpha(\rho, \pi) \leq 1$ and the larger the ratio the better the policy. In the following, we primarily use the approximation ratio as a secondary metric to understand the convergence behavior of an RLHF method, i.e., when the number of samples is sufficiently large.

### 3.1 Existing RLHF is not Strategyproof

Unsurprisingly, we find that existing RLHF algorithms are not strategyproof. Exemplarily, we consider two recently proposed RLHF methods for learning from diverse human preferences [37, 7]. Whereas Zhong et al. [37] aims to maximize social welfare like we do, Chakraborty et al. [7] consider a maximin objective, that is, they wish to maximize the worst-case utility across all labelers. While this is different from the social welfare objective that we consider, it does not prevent us from analyzing the strategyproofness of their algorithm or lack thereof.

**Proposition 3.3.** *Existing RLHF methods such as Pessimistic Social Welfare [37] and MaxMin-RLHF [7] are not strategyproof.*

Next, we wish to understand what consequences being manipulable has on the policy performance of the RLHF algorithm. After all, one could imagine failing to guarantee strategyproofness but still learning a nearly optimal policy. This is in general not the case and we show that the performance can degrade arbitrarily in the worst-case even if only a single labeler is strategic. We show this at the example of the Pessimistic Social Welfare approach from Zhong et al. [37].

**Proposition 3.4.** *Let at least one out of the $k$ labelers report strategically. Let $\hat{\pi}$ denote the output of the Pessimistic Social Welfare algorithm [37]. Recall that $\|\theta\|_2 \leq B$ and $\|\phi(s, a)\|_2 \leq L$. In the worst-case, for $n$ sufficiently large, the social welfare of $\hat{\pi}$ is upper bounded as $\mathcal{W}(\hat{\pi}) \leq \varepsilon$, whereas the optimal social welfare is at least $\mathcal{W}(\pi^*) \geq BL - 2\varepsilon$ for any $\varepsilon > 0$. Hence, the suboptimality of Pessimistic Social Welfare is lower bounded by $\mathrm{SubOpt}(\hat{\pi}) \geq BL - 3\varepsilon$.*

*In other words, the policy learned by Pessimistic Social Welfare can be almost arbitrarily bad.*

*Proof Sketch.* We provide a simple example for a contextual bandit where the first labeler strongly disagrees with all other labelers, but can exert significant influence on the computed policy by overstating its preference in a dimension of the features that is otherwise irrelevant to all labelers' utility (i.e., $\theta_i^*$ is zero in said dimension for all labelers). □

### 3.2 Inherent Limitations of Strategyproof RLHF

We have seen that existing RLHF approaches are not strategyproof, but can be manipulated by labelers to the detriment of policy alignment with social welfare. We now also show that any RLHF algorithm that satisfies strategyproofness must suffer at least constant suboptimality (irrespective of the number of samples or policy coverage) and has an approximation ratio of at most $1/k$. We thereby face a fundamental trade-off between incentive alignment (strategyproofness) and policy alignment (social welfare maximization) in RLHF with strategic preference labeling.

**Theorem 3.5.** *The output $\hat{\pi}$ of any strategyproof RLHF algorithm has worst-case expected suboptimality at least $\mathrm{SubOpt}(\hat{\pi}) \geq \frac{k-1}{k}$, where $k$ denotes the number of labelers.*

*Proof Sketch.* We can map each RLHF instance to a voting problem and map $\hat{\pi}$ to a decision rule $f$ for the latter, such that $f$ always outputs the same alternative (or distribution of alternatives) as $\hat{\pi}$ does. This construction ensures that if $\hat{\pi}$ is stratgyproof, then $f$ is, too. The Gibbard–Satterthwaite theorem [16, 31] says that any strategyproof rule must be either a dictatorial rule or a "duple", i.e., either it always selects the most preferred alternative of a fixed voter, or selects among a fixed pair of alternatives. Hence, if $\hat{\pi}$ is strategyproof, it must behave either as a dictatorial rule, always selecting the most preferred action of a fixed labeler, or as a duple, always selecting the outcome among a fixed pair of actions. The former case leads to low social welfare values for instances in which all the other labelers' rewards are negatively correlated with that of the fixed labeler. The latter leads to low social

welfare values for instances in which the fixed pair of actions have almost zero value to all labelers. In both cases, the suboptimality gaps are at least $(k-1)/k$. □

Theorem 3.5 implies that even with infinitely many samples, no strategyproof RLHF algorithm converges to the optimal policy in the worst case. This is also reflected in the following upper bound on the multiplicative approximation ratio of any strategyproof algorithm.

**Corollary 3.6.** *The approximation ratio of any strategyproof RLHF method is* $\alpha(\rho, \hat{\pi}) \leq \frac{1}{k}$.

In other words, any strategyproof RLHF algorithm achieves $k$-times worse social welfare compared to the optimal policy in the worst case.

## 4 Approximate Strategyproofness: Pessimistic Median of MLEs

We first consider the contextual bandit problem and discuss the extension to MDPs in Section 5. Our previous Theorem 3.5 suggests that without additional assumptions about the problem instance, we cannot reconcile strategyproofness with social welfare maximization. For this reason, we here introduce an additional assumption about the structure of the initial state distribution (i.e., context distribution) and the policy space.

**Assumption 2.** The set $\{\mathbb{E}_{s\sim\rho}[\phi(s, \pi(s))] : \pi \in \Pi\}$ spans a hyperrectangle in $\mathbb{R}^d$.

Specifically, in the simplest case when $\mathbb{E}_{s\sim\rho}[\phi(s, \pi(s))] \in [-1, 1]^d$, this means that for any $\boldsymbol{z} \in [-1, 1]^d$ there exists $\pi \in \Pi$ such that $\|\mathbb{E}_{s\sim\rho}[\phi(s, \pi(s)) - \boldsymbol{z}]\|_2 = 0$.[5]

We propose to use a median rule over learned reward parameters in combination with pessimistic estimates to achieve approximate strategyproofness while maximizing social welfare. To do so, we must first introduce a few key concepts and quantities.

**MLEs and Confidences.** Let $\mathcal{D}_i = (s^{i,j}, a_0^{i,j}, a_1^{i,j}, o^{i,j})_{1\leq j\leq n}$ be the preference data reported by labeler $i \in [k]$ where $o^{i,j} \sim \mathbb{P}_{\theta_i}(\cdot \mid s^{i,j}, a_0^{i,j}, a_1^{i,j})$ is sampled from a BT model w.r.t. some (a priori) unknown and potentially manipulated reward parameter $\theta_i$. Given the observations $\mathcal{D}_i$, the Maximum Likelihood Estimate (MLE) of $\theta_i$ is the maximizer of the log-likelihood

$$\hat{\theta}_i^{\mathrm{MLE}} \in \operatorname*{argmax}_\theta \sum_{j=1}^n \log \mathbb{P}_\theta\big(o^{i,j} \mid s^{i,j}, a_0^{i,j}, a_1^{i,j}\big).$$

We wish to establish confidences around the MLE. To this end, let $x^{i,j} = \phi(s^{i,j}, a_0^{i,j}) - \phi(s^{i,j}, a_1^{i,j})$ and consider the covariance matrix $\Sigma_{\mathcal{D}_i} = \frac{1}{n}\sum_{j=1}^n x^{i,j}(x^{i,j})^\top$. For convenience, we here assume that $\Sigma_{\mathcal{D}_i}$ is positive definite. Otherwise, we can always consider $\Sigma_{\mathcal{D}_i} + \lambda_i I$ for $\lambda_i > 0$, which has a negligible effect on our results when choosing $\lambda_i$ of order $\frac{d+\log(1/\delta)}{n}$ (see, e.g., [38]). The confidence ellipsoid around $\hat{\theta}_i^{\mathrm{MLE}}$ is then given by

$$C_i := \{\theta \in \mathbb{R}^d \colon \|\hat{\theta}_i^{\mathrm{MLE}} - \theta\|_{\Sigma_{\mathcal{D}_i}} \leq f(d, n, \delta)\}.$$

It is well-known that when choosing $f(d, n, \delta) \approx \sqrt{\frac{d+\log(k/\delta)}{n}}$, it holds with probability at least $1 - \delta$ that $\theta_i \in C_i$ (see Appendix A.5 for details).

**Pessimistic Median Return.** A fundamental insight from social choice theory is that under certain conditions aggregating preferences according to a median rule is strategyproof, such as in resource allocation in one dimension [26]. However, in our case, the *high-dimensionality* of features and reward parameters, the *uncertainty* about rewards, and the *policy optimization* pose additional unique challenges that can cause a median rule to become manipulable by the labelers.

To incorporate our uncertainty about the reward parameters, we consider the *pessimistic median return* of a policy defined as the return of a policy w.r.t. the worst-case *coordinate-wise median* over confidence sets $C_1, \ldots, C_k$. In other words, we consider the worst-case performance of policies $\pi$ with respect to $\mathrm{med}(\theta_1, \ldots, \theta_k)$, where med denotes the coordinate-wise median and $\theta_i$ is element in $C_i$. We outline the Pessimistic Median of MLEs approach in Algorithm 1.

---

[5]Without much additional difficulty we can relax this to $\|\mathbb{E}_{s\sim\rho}[\phi(s, \pi(s)) - \boldsymbol{z}]\|_2 \leq \varepsilon$ for some $\varepsilon > 0$ at the cost of additive expressions of order $\varepsilon$ in our results.

---
**Algorithm 1** Pessimistic Median of MLEs (Pessimistic MoMLEs)
---
1: **input** offline preference data $\mathcal{D} = (\mathcal{D}_1, \ldots, \mathcal{D}_k)$
2: **for** every labeler $i \in [k]$ **do**
3:     compute the MLE $\hat{\theta}_i^{\text{MLE}}$ from $\mathcal{D}_i$
4:     construct confidence set $C_i := \{\theta \in \mathbb{R}^d : \|\hat{\theta}_i^{\text{MLE}} - \theta\|_{\Sigma_{\mathcal{D}_i}} \leq f(d, n, \delta)\}$
5: **end for**
6: get the median confidence set $\mathscr{C} := \{\text{med}(\theta_1, \ldots, \theta_k) : \theta_i \in C_i \text{ for } i \in [k]\}$
7: compute the pessimistic median return w.r.t. $\mathscr{C}$ given by

$$\underline{\mathcal{W}}(\pi) := \min_{\theta \in \mathscr{C}} \mathbb{E}_{s \sim \rho} \left[ \langle \theta, \phi(s, \pi(s)) \rangle \right]$$

8: **return** $\hat{\pi}(\mathcal{D}) = \text{argmax}_{\pi \in \Pi} \underline{\mathcal{W}}(\pi)$
---

## 4.1 Approximate Strategyproofness

We begin the analysis by showing that the Pessimistic Median of MLEs is approximately strategyproof. Perhaps surprisingly, the degree up to which the algorithm is strategyproof depends on the *uniform policy coverage* of every labeler's data. We discuss this in more detail further below.

**Theorem 4.1.** *Pessimistic Median of MLEs is $\tilde{\mathcal{O}}(\kappa_i \sqrt{d/n})$-strategyproof for labeler $i$, where $\kappa_i := \max_{\pi \in \Pi} \|\mathbb{E}_{s \sim \rho}[\phi(s, \pi(s))]\|_{\Sigma_{\mathcal{D}_i}^{-1}}$ is the uniform policy coverage of $\mathcal{D}_i$.[6]*

*More precisely, for every labeler $i \in [k]$, any other labelers' reports $\mathcal{D}_{-i}$ and deviation $\tilde{\theta}_i \neq \theta_i^*$, with probability at least $1 - \delta$, the gain from misreporting is upper bounded as*

$$J_i\big(\hat{\pi}(\tilde{\mathcal{D}}_i, \mathcal{D}_{-i})\big) - J_i\big(\hat{\pi}(\mathcal{D}_i^*, \mathcal{D}_{-i})\big) \leq const \cdot \kappa_i \sqrt{\frac{d + \log(k/\delta)}{n}},$$

*where the labels in $\tilde{\mathcal{D}}_i$ are sampled from $\mathbb{P}_{\tilde{\theta}_i}$ and the labels in $\mathcal{D}_i^*$ are truthfully sampled from $\mathbb{P}_{\theta_i^*}$.*

*Proof Sketch.* The key challenge is that the estimation errors of the reward parameters may unintentionally alter the median computation and thereby create unintended incentives for misreporting. To bound the gain from misreporting, we analyze the effect of estimation errors in conjunction with deviating choices of $\theta_i$ on the learned policy. Using concentration inequalities, we show that the deviation in each labeler's expected return is proportional to the estimation error, which scales as $\sqrt{d/n}$. The worst-case impact on strategyproofness is then controlled by the uniform coverage coefficient $\kappa_i$, which measures how well the labeler's data constrains policy choices. $\square$

Whereas the $\sqrt{d/n}$ factor may be expected due to the construction of the confidence ellipsoids of corresponding size, the dependence on the *uniform* policy coverage coefficient $\kappa_i$ is unexpected at first, since coverage of only the optimal policy $\pi^*$ is usually sufficient in offline RL [29, 12, 36, 38]. However, in our case, we are not bounding the suboptimality of a learned policy but rather analyzing the strategic incentives of labelers. This shifts the focus to the range of possible policies that could result from different labeler behavior. Since labelers can, in principle, report arbitrarily misleading reward parameters—potentially inducing policies far from $\pi^*$—bounding their incentive to deviate requires uniform policy coverage rather than coverage of any single specific policy. This ensures that no matter what policy is induced by a misreport, the confidence set remains well-constrained and bounds the potential gain from misreporting.

## 4.2 Social Welfare Maximization

We have shown that being truthful is approximately optimal for all labelers. Next, we provide guarantees on the suboptimality and the approximation ratio of the Pessimistic Median of MLEs algorithm when the labelers are either truthful or act according to their (potentially manipulating)

---

[6]Note that for any positive definite matrix $\Sigma$ and vector $x$, we can write $\|x\|_{\Sigma^{-1}} = \|\Sigma^{-1/2}x\|_2$. It is also worth noting that labeler $i$ cannot influence the coverage coefficient $\kappa_i$ as it only depends on the state-action pairs and not the preference labels. Hence, labeler $i$ has no influence on the incentive strength of the algorithm.

weakly dominant strategy, which we show to exist. We begin with the case when the labelers are truthful, which is a $\tilde{\mathcal{O}}(\kappa_i\sqrt{d/n})$-dominant strategy according to our previous Theorem 4.1.

**Theorem 4.2.** *Let $\hat{\pi}$ be the output of the Pessimistic Median of MLEs algorithm and suppose that all labelers report truthfully. With probability at least $1 - \delta$:*

$$\text{SubOpt}(\hat{\pi}) \leq const \cdot \left( \sqrt{\frac{d\log(k/\delta)}{k}} + \max_{i \in [k]} \kappa_i^* \cdot k \sqrt{\frac{d + \log(k/\delta)}{n}} \right) \tag{2}$$

*where $\kappa_i^* := \|\mathbb{E}_{s\sim\rho}[\phi(s, \pi^*(s))]\|_{\Sigma_{\mathcal{D}_i}^{-1}}$ is the optimal policy coverage of labeler $i$.*

*Proof Sketch.* The suboptimality arises from two sources: (1) the deviation of the pessimistic median from the average, and (2) the deviation of each true reward parameter from its worst-case estimate in its respective confidence set. The first term follows from median concentration around the mean, contributing an error of $\mathcal{O}(\sqrt{d\log(k/\delta)/k})$. The second term is upper bounded by $\mathcal{O}(\sqrt{(d + \log(k/\delta))/n})$, scaled by the worst-case policy coverage coefficient. Here, taking the median over confidence sets introduces an additional factor of $k$. □

We also show that the Pessimistic Median of MLEs algorithm enjoys a suboptimality upper bound matching the one from Theorem 4.2 under any weakly dominant strategy it induces.

**Proposition 4.3.** *When the labelers report their preferences according to any weakly dominant strategy under Pessimistic Median of MLEs, with probability at least $1 - \delta$, the output $\hat{\pi}$ satisfies:*

$$\text{SubOpt}(\hat{\pi}) \leq const \cdot \left( \sqrt{\frac{d\log(k/\delta)}{k}} + \max_{i \in [k]} \kappa_i^* \cdot k \sqrt{\frac{d + \log(k/\delta)}{n}} \right).$$

*where $\kappa_i^* := \|\mathbb{E}_{s\sim\rho}[\phi(s, \pi^*(s))]\|_{\Sigma_{\mathcal{D}_i}^{-1}}$ is the optimal policy coverage of labeler $i$.*

The bounds in Theorem 4.2 and Proposition 4.3 suggest two sources of suboptimality. The first term stems from approximating the social welfare function using the coordinate-wise median, which improves as the number of labelers increases. The second term results from the estimation of the underlying reward parameters, where the use of a median rule introduces an additional factor of $k$. Overall, as the number of samples increases and as the number of labelers grows, the Pessimistic Median of MLEs algorithm converges to the optimal policy.

**Remark 4.4** (Suboptimality Lower Bound). *The worst-case suboptimality of any RLHF algorithm in our problem setup is lower bounded by $\Omega(\sqrt{d/n})$. This can be derived using a similar worst-case problem instance construction to the one in Zhu et al. [38].*

We want to highlight the performance bounds of the Pessimistic Median of MLEs algorithm in two interesting special cases: (1) when there is only a single labeler so that $k = 1$, and (2) when all $k$ labelers have identical reward functions.

**Corollary 4.5.** *When there is only a single labeler, with probability at least $1 - \delta$:*

$$\text{SubOpt}(\hat{\pi}) \leq const \cdot \kappa_1^* \sqrt{\frac{d + \log(k/\delta)}{n}}.$$

*When all $k$ labelers have the same reward function, with probability at least $1 - \delta$:*

$$\text{SubOpt}(\hat{\pi}) \leq const \cdot \max_{i \in [k]} \kappa_i^* \cdot k \sqrt{\frac{d + \log(k/\delta)}{n}}.$$

The result for the single labeler matches the existing bounds in the offline RLHF literature and is tight up to constants. Interestingly, we observe that in the special case of $k$ labelers with identical reward functions, the Pessimistic Median of MLEs avoids the additive $\mathcal{O}(\sqrt{d\log(k/\delta)/k})$ suboptimality but still suffers from an additional factor of $k$ as the algorithm anticipates strategic manipulation and preemptively takes the median over the confidence sets.

Finally, we can also derive a lower bound on the approximation ratio of Algorithm 1.

**Corollary 4.6.** *Suppose $\mathcal{W}(\pi^*) > 0$ is constant. When the number of samples is sufficiently large and provide sufficient coverage of the optimal policy, with probability at least $1 - \delta$, the approximation ratio of the Pessimistic Median of MLEs algorithm is given by $\alpha(\rho, \hat{\pi}) \geq 1 - \mathcal{O}\left(\sqrt{d\log(k/\delta)/k}\right)$.*

# 5 Extension to Markov Decision Processes

We now extend our algorithm and our previous results to MDPs. Recall that we consider trajectory-wise preferences so that labeler $i$ provides a preferences $o^{i,j}$ over two trajectories $\tau_0^{i,j}$ and $\tau_1^{i,j}$ given initial state $s^{i,j}$ according to a BT model $\mathbb{P}_{\theta_i}$ as defined in Section 3. Like before, the MLE of $\theta_i$ is given by the maximizer of the log-likelihood

$$\theta_i^{\text{MLE}} := \underset{\theta}{\operatorname{argmax}} \sum_{j=1}^{n} \log \mathbb{P}_\theta(o^{i,j} \mid s^{i,j}, \tau_0^{i,j}, \tau_1^{i,j}).$$

To construct the confidence ellipsoid around the MLE, let $x^{i,j} = \sum_{h=1}^{H}(\phi(s_h^{i,j}, a_h^{i,j}) - \phi(\bar{s}_h^{i,j}, \bar{a}_h^{i,j}))$ with $s_1^{i,j} = \bar{s}_1^{i,j} = s^{i,j}$ and consider the adapted covariance matrix $\Sigma_{\mathcal{D}_i} = \sum_{j=1}^{n} x^{i,j}(x^{i,j})^\top$. Note that this agrees with our previous definition in the contextual bandit when $H = 1$.

To derive the pessimistic estimate of the median social welfare, we now consider the state occupancy of a policy $\pi$ given by $q_\pi(s \mid \rho) := \frac{1}{H} \sum_{h=1}^{H} \mathcal{P}_h(s_h = s \mid \rho, \pi)$. We can then express the expected return of policy $\pi$ w.r.t. reward parameter $\theta$ as $\mathbb{E}_{s \sim \rho}[V_\theta^\pi(s)] = \mathbb{E}_{s \sim q_\pi}[\langle \theta, \phi(s, \pi(s)) \rangle]$ and the pessimistic estimate of the median social welfare is given by $\underline{\mathcal{W}}(\pi) := \min_{\theta \in \mathscr{C}} \mathbb{E}_{s \sim q_\pi}[\langle \theta, \phi(s, \pi(s)) \rangle]$. The remainder of the Pessimistic Median of MLEs algorithm proceeds the same.

We assume the analogue of Assumption 2 for MDPs.

**Assumption 3.** The set $\{\mathbb{E}_{s \sim q_\pi}[\phi(s, \pi(s))] : \pi \in \Pi\}$ spans a hyperrectangle in $\mathbb{R}^d$.

Under Assumption 3, we obtain the following extension of Theorem 4.1 that shows the approximate strategyproofness of the Pessimistic Median of MLEs.

**Theorem 5.1.** *Pessimistic Median of MLEs is $\tilde{\mathcal{O}}(\nu_i \sqrt{d/n})$-strategyproof for labeler $i$ with uniform policy coverage coefficient $\nu_i := \max_{\pi \in \Pi} \|\mathbb{E}_{s \sim q_\pi}[\phi(s, \pi(s))]\|_{\Sigma_{\mathcal{D}_i}^{-1}}$.*

*More precisely, for every labeler $i \in [k]$, any other labelers' data $\mathcal{D}_{-i}$ and manipulated reward parameter $\tilde{\theta}_i \neq \theta_i^*$, with probability at least $1 - \delta$, the gain from misreporting is bounded as*

$$J_i\big(\hat{\pi}(\tilde{\mathcal{D}}_i, \mathcal{D}_{-i})\big) - J_i\big(\hat{\pi}(\mathcal{D}_i^*, \mathcal{D}_{-i})\big) \leq const \cdot \nu_i, \sqrt{\frac{d + \log(k/\delta)}{n}}.$$

*where the labels in $\tilde{\mathcal{D}}_i$ are sampled from $\mathbb{P}_{\tilde{\theta}_i}$ and the labels in $\mathcal{D}_i^*$ are truthfully sampled from $\mathbb{P}_{\theta_i^*}$.*

The suboptimality upper bounds under truthful or weakly dominant reporting also take a similar form to their counterparts in Theorem 4.2 and Proposition 4.3. Similarly to before, the coverage of the optimal policy is enough.

**Theorem 5.2.** *When all labelers report truthfully or report according to their weakly dominant strategies, then with probability at least $1 - \delta$:*

$$\operatorname{SubOpt}(\hat{\pi}) \leq const \cdot \left( \sqrt{\frac{d \log(k/\delta)}{k}} + \max_{i \in [k]} \nu_i^* \cdot k \sqrt{\frac{d + \log(k/\delta)}{n}} \right)$$

*where $\nu_i^* := \|\mathbb{E}_{s \sim q_{\pi^*}}[\phi(s, \pi^*(s))]\|_{\Sigma_{\mathcal{D}_i}^{-1}}$.*

Note that we can also extend the corollaries from Section 4 to MDPs in a similar fashion.

# 6 Discussion

We studied how to robustify offline RLHF against strategic preference labeling in a pluralistic alignment setting with multiple labelers. We demonstrated a fundamental trade-off between incentive alignment and policy alignment and proposed the Pessimistic Median of MLEs algorithm that is based on pessimistic estimates of the median return of a policy. We showed that this algorithm is $\tilde{\mathcal{O}}(\sqrt{d/n})$-strategyproof while guaranteeing suboptimality of at most $\tilde{\mathcal{O}}(\sqrt{d/k} + k\sqrt{d/n})$. There are many directions for future work. It will be interesting to study strategyproofness for non-linear reward functions, parameterized or otherwise restricted policy classes, as well as more general preference models to the BT model used here. Another interesting future direction is to empirically evaluate the effect of strategic preference labeling on AI alignment and to validate algorithmic mechanisms designed to mitigate strategic manipulation. To do so at scale, e.g., in the context of LLM fine-tuning, we can expect scalability to come at the cost of theoretical guarantees.

## Acknowledgments and Disclosure of Funding

This work is supported by the EPSRC Prosperity Partnership FAIR (grant number EP/V056883/1), and by the ETH AI Center through an ETH AI Center Postdoctoral Fellowship to TKB. MK receives funding from the ERC under the European Union's Horizon 2020 research and innovation programme FUN2MODEL (grant agreement No. 834115).

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

# A Proofs

## A.1 Proof of Proposition 3.3

**Proposition 3.3.** *Existing RLHF methods such as Pessimistic Social Welfare [37] and MaxMin-RLHF [7] are not strategyproof.*

*Proof.* We construct straightforward examples for each of the algorithms and show that the algorithms are not strategyproof. W.l.o.g. we assume that $n$ is sufficiently large with appropriate policy coverage so that the algorithms are able to obtain perfect estimates. We thereby also avoid re-defining the preference model $\mathbb{P}_\theta$ to fit the models considered in the respective work. W.l.o.g. we also ignore the KL-regularization w.r.t. the reference policy $\pi_{\text{ref}}$ in MaxMin-RLHF, which would only add notational burden. In the following examples, the horizon is set to $H = 1$, that is, we consider contextual bandit problems.

**Pessimistic Social Welfare:** Suppose there are two labelers with true reward parameters $\theta_1^* = (1,0)$ and $\theta_2^* = (0,1)$. Moreover, suppose that for all $s$, we have $\phi(s,a) = (1/2, 1/2)$ and $\phi(s,b) = (3/4, 0)$. If both labelers report truthfully, Pessimistic Social Welfare reports a policy $\pi(s) = a$ for all $s$ as action $a$ is maximizing social welfare. In this case, labeler 1 receives utility $1/2$. Suppose that labeler 1 misreports as $\tilde{\theta}_1 = (1,-1)$. As a result, the social welfare maximizing policy is $\pi(s) = b$, which yields utility $3/4$ for labeler 1. Hence, misreporting is beneficial to labeler 1 and Pessimistic Social Welfare not strategyproof.

**MaxMin-RLHF:** Consider the simple example where $\theta_1^* = (1,0)$ and $\theta_2^* = (1/2, 1/2)$ as well as $\phi(s,a) = (1/2, 1/2)$ and $\phi(s,b) = (3/4, 0)$ for all $s$. If both labelers report truthfully, the MaxMin-RLHF would compute the policy $\pi(s) = a$ which yields a return of $1/2$ for both labelers. However, suppose labeler 1 reports $\tilde{\theta}_1 = (1,-1)$ while labeler 2 truthfully reports $\theta_2^* = (1/2, 1/2)$. In this case, MaxMin-RLHF returns a policy $\tilde{\pi}(s) = b$ as this maximizes the minimal utility w.r.t. the reported reward parameters. The return for labeler 1 under policy $\tilde{\pi}$ is $3/4$, which means that misreporting is to the benefit of labeler 1 and MaxMin-RLHF not strategyproof. $\qquad\square$

## A.2 Proof of Proposition 3.4

**Proposition 3.4.** *Let at least one out of the $k$ labelers report strategically. Let $\hat{\pi}$ denote the output of the Pessimistic Social Welfare algorithm [37]. Recall that $\|\theta\|_2 \leq B$ and $\|\phi(s,a)\|_2 \leq L$. In the worst-case, for $n$ sufficiently large, the social welfare of $\hat{\pi}$ is upper bounded as $\mathcal{W}(\hat{\pi}) \leq \varepsilon$, whereas the optimal social welfare is at least $\mathcal{W}(\pi^*) \geq BL - 2\varepsilon$ for any $\varepsilon > 0$. Hence, the suboptimality of Pessimistic Social Welfare is lower bounded by $\mathrm{SubOpt}(\hat{\pi}) \geq BL - 3\varepsilon$.*

*In other words, the policy learned by Pessimistic Social Welfare can be almost arbitrarily bad.*

*Proof.* We construct a contextual bandit problem, where Pessimistic Social Welfare is arbitrarily bad if even a single labeler is strategic. We here assume that Pessimistic Social Welfare receives infinitely many samples from each labeler with full policy coverage.

Let $\theta_1^* = (0,1,0)$ and $\theta_i^* = (\frac{B}{k-1}, 0, 0)$ for all $i \neq 1$. Moreover, suppose that $\phi(s,a) = (\sqrt{L^2 - 2\varepsilon^2}, 0, 0)$ and $\phi(s,b) = (0, \varepsilon, \sqrt{L^2 - \varepsilon^2})$ for all $s$. Under truthful reporting, Pessimistic Social Welfare computes the policy $\pi(s) = a$, which clearly maximizes social welfare. In particular, the optimal social welfare is given by $\mathcal{W}(\pi^*) = B\sqrt{L^2 - 2\varepsilon^2} \geq B(L - \sqrt{2}\varepsilon)$. In this case, labeler 1 receives utility zero. However, suppose that labeler 1 misreports its reward parameter as $\tilde{\theta}_1^* = (0,0,B)$. In this case, Pessimistic Social Welfare returns the policy $\tilde{\pi}(s) = b$, which has social welfare $\mathcal{W}(\tilde{\pi}) = \varepsilon$, whereas the utility of labeler 1 is $\varepsilon$ which is the best possible outcome for labeler 1.

As a result, even if only labeler 1 misreports, then $\mathcal{W}(\hat{\pi}) = \varepsilon$ and the suboptimality is at least $\mathrm{SubOpt}(\hat{\pi}) = \mathcal{W}(\pi^*) - \varepsilon \geq BL - 3\varepsilon$. We can choose $\varepsilon > 0$ arbitrarily small (in particular, note that $\varepsilon$ does not depend on the reward parameter so that any estimation error would have no effect). This means that the suboptimality of Pessimistic Social Welfare can be maximal. $\qquad\square$

## A.3 Proof of Theorem 3.5

**Theorem 3.5.** *The output $\hat{\pi}$ of any strategyproof RLHF algorithm has worst-case expected suboptimality at least* $\mathrm{SubOpt}(\hat{\pi}) \geq \frac{k-1}{k}$, *where $k$ denotes the number of labelers.*

*Proof.* Suppose for the sake of contradiction that the (worst-case) suboptimality gap of $\hat{\pi}$ is $\frac{k-1}{k} - \epsilon$ for some $\epsilon > 0$. We show that if this were true, we could construct a voting rule based on $\hat{\pi}$ that is *strategyproof*, *non-dictatorial*, and *onto at least three alternatives*, contradicting the Gibbard–Satterthwaite theorem [16, 31]. The Gibbard–Satterthwaite theorem asserts that such a voting rule does not exist. We will consider the case where $\hat{\pi}$ is deterministic and discuss how the proof generalizes to the case with randomized $\hat{\pi}$, too.

Specifically, consider a voting instance with $k$ voters and $m$ alternatives $a_1, \ldots, a_m$. A voting rule $f$ maps every possible preference profile of the voters to one of the alternatives.

To construct a voting rule based on $\hat{\pi}$, we first map every voting instance $I_V = \prec := (\prec_1, \ldots, \prec_k)$ to an RLHF instance $I_{\mathrm{RLHF}}$ as follows.

- Let there be one state (so we omit the state in what follows) and $m$ actions $a_1, \ldots, a_m$, each corresponding to an alternative in $I_V$.

- The feature embedding $\phi$ maps each action $a_\ell$ to the unit vector whose $\ell$-th component is 1.

- Let there be $k$ labelers, each corresponding to a voter in $I_V$. Each labeler $i$'s parameter $\theta_i^*$ is a vector in which the $\ell$-th component is defined as follows:

    - 1 if $a_\ell$ is the most preferred alternative according to voter $i$'s preference order $\prec_i$ in $I_V$.

    - $1 - \delta$ (for a sufficiently small $\delta$) if $a_\ell$ is the second most preferred alternative according to $\prec_i$.

    - $\delta \cdot (m - j)$ if $a_\ell$ is the $j$-th most preferred alternative, $j > 2$, according to $\prec_i$.

    The parameters ensure that each labeler $i$'s preference over the actions is the same as $\prec_i$.

With the above map from $I_V$ to $I_{\mathrm{RLHF}}$, we then let $f$ be a voting rule that outputs alternative $a_\ell$ if $\hat{\pi}$ outputs action $a_\ell$ in $I_{\mathrm{RLHF}}$. Clearly, $f$ must be strategyproof given our assumption that $\hat{\pi}$ is strategyproof and the fact that $I_{\mathrm{RLHF}}$ preserves the preference orders in $I_V$. We next argue that, given the assumption that $\hat{\pi}$ has a suboptimality gap of at most $\frac{k-1}{k} - \epsilon$, $f$ must be non-dictatorial and onto at least three alternatives.

**$f$ is Non-Dictatorial.** Suppose that $f$ is dictatorial; say, it always outputs the most preferred alternative of voter 1. This means that $\hat{\pi}$ must output $a_1$ on the RLHF instance that the following $I_V = \prec$ instance maps to:

$$a_1 \prec_1 a_2 \prec_1 \cdots \prec_1 a_{m-1} \prec_1 a_m$$
$$a_2 \prec_i a_3 \prec_i \cdots \prec_i a_m \quad \prec_i a_1 \quad \text{for all } i = 2, \ldots, k.$$

However, this contradicts the assumption that the suboptimality gap of $\hat{\pi}$ is at most $\frac{k-1}{k} - \epsilon$. To see this, note that $a_1$ achieves social welfare 1 in $I_{\mathrm{RLHF}}$, while $a_2$ achieves social welfare $k(1 - \delta)$. The suboptimality is then at least $\frac{k-1-\delta k}{k-\delta k} > \frac{k-1}{k} - \epsilon$ when $\delta \to 0$.

**$f$ is Onto.** Similarly, we can argue that $f$ must be onto at least three alternatives by considering the set of all possible voting instances where all voters' preferences are identical. In this case, $\hat{\pi}$ must always output either the most or the second preferred actions of the labelers in the corresponding RLHF instances; otherwise, the suboptimality gap of $\hat{\pi}$ can be arbitrarily close to 1 when $\delta \to 0$. Consequently, when all possible preference orders are considered, $\hat{\pi}$, and hence $f$, must be onto at least three different alternatives (suppose that $k \geq 4$).

As a result, we obtain a voting rule that is strategyproof, non-dictatorial, and onto at least three alternatives. This contradicts the Gibbard–Satterthwaite theorem. The stated lower bound on the suboptimality gap then follows for deterministic policies.

**Randomized Policies.** To further argue that the same bound holds even when $\hat{\pi}$ is randomized, we invoke a generalization of the Gibbard–Satterthwaite theorem to randomized voting rules [17], which states that a voting rule, if strategyproof, must be a probability mixture of dictatorial rules and "duples". A voting rule is a duple if it restricts its outcomes, over all possible instances, to a fixed pair of alternatives.

Similarly to our approach above, we construct a voting rule $f$ based on $\hat{\pi}$ the same way we did above and argue that, if $\hat{\pi}$ is strategyproof and has a suboptimality gap of $\frac{k-1}{k} - \epsilon$ for some $\epsilon > 0$, then $f$ *cannot* be a probability mixture of dictatorial rules and duples.

Suppose for the sake of contradiction that $f$ is a mixture of dictatorial rules and duples. Consider the following set of voting instances $(\prec^j)_j$, each involving a set of alternatives $\{a_1, \ldots, a_k, b_1, \ldots, b_K\}$, $K \geq 4/\epsilon$ (hence, $m = k + K$). In each instance $\prec^j = (\prec_1^j, \ldots, \prec_k^j)$, the preference order $\prec_i^j$ ranks $a_i$ first, $b_j$ second, and all other alternatives according to the order $a_1, \ldots, a_k, b_1, \ldots, b_k$.

By assumption, $f$ is a mixture of dictatorial rules and duples. Now that there are more than $K$ alternatives while each duple selects at most two alternatives, there must be at least one alternative among $b_1, \ldots, b_K$ that is selected by the duples with probability at most $2/K$. W.l.o.g., let this alternative be $b_1$ and consider the instance $\prec^1$, in which each voter $i$ ranks $a_i$ the first and $b_1$ the second.

Clearly, in this instance, the policy that outputs $b_1$ deterministically achieves social welfare $k(1 - \delta)$. Moreover, any other alternatives yields a social welfare of at most $1 + \delta m k$, as each of these alternatives is ranked first by at most one labeler and third or even lower by all other labelers. Since $f$ selects $b_1$ with probability at most $2/K$, so does $\hat{\pi}$. This means that the social welfare achieved by $\hat{\pi}$ is at most

$$\frac{2}{K} \cdot k(1 - \delta) + \left(1 - \frac{2}{K}\right)(1 + \delta m k) < \epsilon k/2 + 1 + \delta m k.$$

This gives a suboptimality gap of $1 - \frac{\epsilon k/2 + 1 + \delta m k}{k(1-\delta)} > \frac{k-1}{k} - \epsilon$ when $\delta \to 0$. $\qquad\square$

## A.4 Proof of Corollary 3.6

**Corollary 3.6.** *The approximation ratio of any strategyproof RLHF method is $\alpha(\rho, \hat{\pi}) \leq \frac{1}{k}$.*

*Proof.* This is a direct consequence of Theorem 3.5. $\qquad\square$

## A.5 Proof of Theorem 4.1

Before we begin with some preliminaries that we will repeatedly use in the proofs of both Theorem 4.1 and Theorem 4.2. Firstly, we recall a standard MLE concentration bound, which can be found, for instance, in [38].

**Lemma A.1** (MLE Concentration Bound). *In the contextual bandit problem, with probability at least $1 - \delta$,*

$$\|\hat{\theta}_i^{\text{MLE}} - \theta_i^*\|_{\Sigma_{\mathcal{D}_i}} \leq const \cdot \sqrt{\frac{d + \log(1/\delta)}{\gamma^2 n}},$$

*where $\gamma := 1/(2 + \exp(-LB) + \exp(LB))$. Note that we here assume that the covariance matrix $\mathcal{D}_i$ is positive definite. Otherwise, consider $\Sigma_{\mathcal{D}_i} + \lambda I$, which adds an additive term $\lambda B^2$ in the square root.*

*Proof.* See, e.g., [38]. $\qquad\square$

**Lemma A.2** (Median Concentration Bound). *Suppose that $\theta_1, \ldots, \theta_k \in \mathbb{R}^d$ are sampled i.i.d. from some $\sigma$-sub Gaussian distribution. Let $\hat{\theta}_{\text{med}}$ be the coordinate-wise median and $\hat{\theta}_{\text{avg}}$ the average of $\theta_1, \ldots, \theta_k$. Then, for a universal constant $c > 0$, it holds that, for every $t > 0$,*

$$\mathbb{P}\left(\|\hat{\theta}_{\text{med}} - \hat{\theta}_{\text{avg}})\|_2 \geq t\right) \leq 2 \exp\left(-\frac{ckt^2}{d\sigma^2}\right).$$

*Hence, in other words, with probability at least $1 - \delta$:*

$$\|\hat{\theta}_{\text{med}} - \hat{\theta}_{\text{avg}}\|_2 \leq \mathcal{O}\left(\sigma\sqrt{\frac{d\log(1/\delta)}{k}}\right).$$

*Proof.* We begin by proving the median concentration in one dimension. To this end, let $\theta^*_{\text{avg}}$ denote the mean of the distribution. Since each $\theta_i$ is $\sigma$-sub-Gaussian with mean $\theta^*_{\text{avg}}$, the centered variables $X_i = \theta_i - \theta^*_{\text{avg}}$ satisfy

$$\mathbb{P}(|X_i| \geq u) \leq 2\exp\left(-\frac{c_2 u^2}{\sigma^2}\right)$$

for some constant $c_2 > 0$. To control $\mathbb{P}(\hat{\theta}_{\text{med}} \geq \theta^*_{\text{avg}} + t)$, note that if $\hat{\theta}_{\text{med}} \geq \theta^*_{\text{avg}} + t$, at least half of the $\theta_i$ are at least $\theta^*_{\text{avg}} + t$. Define

$$p = \mathbb{P}(\theta_i \geq \theta^* + t) = \mathbb{P}(X_i \geq t).$$

By sub-Gaussianity, $p \leq \exp\left(-\frac{c_2 t^2}{\sigma^2}\right)$. Let $Y = \sum_{i=1}^{k} \mathbf{1}\{\theta_i \geq \theta^* + t\}$, which follows a Binomial$(k, p)$ distribution. Then $\mathbb{P}(\hat{\theta}_{\text{med}} \geq \theta^* + t) \leq \mathbb{P}(Y \geq k/2)$. A Chernoff or Hoeffding bound implies

$$\mathbb{P}(Y \geq k/2) \leq \exp\left(-kD\left(\tfrac{1}{2} \,\|\, p\right)\right),$$

where $D(\tfrac{1}{2} \,\|\, p)$ is the Kullback–Leibler divergence between Bernoulli$(1/2)$ and Bernoulli$(p)$. For $p \ll 1/2$, $D(\tfrac{1}{2} \,\|\, p)$ is bounded below by a constant times $(1/2 - p)^2$. Hence, there exists $c_1 > 0$ such that

$$\mathbb{P}(\hat{\theta}_{\text{med}} \geq \theta^*_{\text{avg}} + t) \leq \exp\left(-\frac{c_1 k t^2}{\sigma^2}\right).$$

By a symmetric argument, $\mathbb{P}(\hat{\theta}_{\text{med}} \leq \theta^*_{\text{avg}} - t) \leq \exp\left(-\frac{c_1 k t^2}{\sigma^2}\right)$. Combining these, we obtain

$$\mathbb{P}(|\hat{\theta}_{\text{med}} - \theta^*_{\text{avg}}| \geq t) \leq \mathbb{P}(\hat{\theta}_{\text{med}} \geq \theta^*_{\text{avg}} + t) + \mathbb{P}(\hat{\theta}_{\text{med}} \leq \theta^*_{\text{avg}} - t) \leq 2\exp\left(-\frac{c_1 k t^2}{\sigma^2}\right).$$

From Hoeffding's inequality we get an analogous bound for $\mathbb{P}(\|\hat{\theta}_{\text{avg}} - \theta^*_{\text{avg}}\| \geq t)$ so that we get the desired result for $d = 1$ using the triangle inequality $|\hat{\theta}_{\text{med}} - \hat{\theta}_{\text{avg}}| \leq |\hat{\theta}_{\text{med}} - \theta^*_{\text{avg}}| + |\theta^*_{\text{avg}} - \hat{\theta}_{\text{avg}}|$.

Finally, this translates to a bound in $d > 1$ dimensions by using Jensen's inequality

$$\|\hat{\theta}_{\text{med}} - \hat{\theta}_{\text{avg}}\|_2 = \sqrt{\sum_{j=1}^{d}\left(\hat{\theta}_{\text{med},j} - \hat{\theta}_{\text{avg},j}\right)^2} \leq \sqrt{d}\max_{j\in[d]}|\hat{\theta}_{\text{med},j} - \hat{\theta}_{\text{avg},j}|$$

and applying the previous bound for each dimension. $\qquad\square$

We are now ready to prove Theorem 4.1

**Theorem 4.1.** *Pessimistic Median of MLEs is $\tilde{\mathcal{O}}(\kappa_i\sqrt{d/n})$-strategyproof for labeler $i$, where $\kappa_i :=$ $\max_{\pi\in\Pi}\|\mathbb{E}_{s\sim\rho}[\phi(s,\pi(s))]\|_{\Sigma_{\mathcal{D}_i}^{-1}}$ is the uniform policy coverage of $\mathcal{D}_i$.[7]*

*More precisely, for every labeler $i \in [k]$, any other labelers' reports $\mathcal{D}_{-i}$ and deviation $\tilde{\theta}_i \neq \theta^*_i$, with probability at least $1 - \delta$, the gain from misreporting is upper bounded as*

$$J_i\left(\hat{\pi}(\tilde{\mathcal{D}}_i, \mathcal{D}_{-i})\right) - J_i\left(\hat{\pi}(\mathcal{D}^*_i, \mathcal{D}_{-i})\right) \leq const \cdot \kappa_i\sqrt{\frac{d + \log(k/\delta)}{n}},$$

*where the labels in $\tilde{\mathcal{D}}_i$ are sampled from $\mathbb{P}_{\tilde{\theta}_i}$ and the labels in $\mathcal{D}^*_i$ are truthfully sampled from $\mathbb{P}_{\theta^*_i}$.*

*Proof.* We begin first with the case where every individual can directly report their reward parameter to the algorithm, hence, removing the noise and uncertainty from the process. In this case, we show that Pessimistic Median of MLEs is exactly strategyproof.

---

[7]Note that for any positive definite matrix $\Sigma$ and vector $x$, we can write $\|x\|_{\Sigma^{-1}} = \|\Sigma^{-1/2}x\|_2$. It is also worth noting that labeler $i$ cannot influence the coverage coefficient $\kappa_i$ as it only depends on the state-action pairs and not the preference labels. Hence, labeler $i$ has no influence on the incentive strength of the algorithm.

**Case 1 (direct access to $\theta_1, \ldots, \theta_k$):** Let us begin with the case where we obtain infinitely many samples with appropriate coverage so that $C_i = \{\theta_i\}$ for all individuals $i \in [k]$. We need to show that reporting $\theta_i^*$ is the optimal strategy for individual $i$ irrespective of the other individuals' strategies.

The following two basic lemmas will prove useful.

**Lemma A.3.** *Let $\theta_{-i} \in \mathbb{R}^{(k-1) \times d}$ be fixed arbitrarily. For any $j \in [d]$ the following holds:*

- *If $\theta_{i,j}^* > 0$ and $\mathrm{med}(\theta_{-i,j}, \theta_{i,j}^*) < 0$, then $\mathrm{med}(\theta_{-i,j}, \theta_{i,j}) < 0$ for all $\theta_i \in \mathbb{R}^d$.*

- *Analogously, if $\theta_{i,j}^* < 0$ and $\mathrm{med}(\theta_{-i,j}, \theta_{i,j}^*) > 0$, then $\mathrm{med}(\theta_{-i,j}, \theta_{i,j}) > 0$ for all $\theta_i \in \mathbb{R}^d$.*

*Proof.* W.l.o.g. let $\theta_{i,j}^* > 0$ and let $\theta_i \in \mathbb{R}^d$. Suppose that $\theta_{i,j} < 0$. It follows directly that $\mathrm{med}(\theta_{-i,j}, \theta_{i,j}) < \mathrm{med}(\theta_{-i,j}, \theta_{i,j}^*)$. Alternatively, suppose that $\theta_{i,j} > 0$. Since $\mathrm{med}(\theta_{-i,j}, \theta_{i,j}^*) < 0$, it means that the median equals some $\theta_{l,j} < 0$ with $l \neq i$. Hence, the median does not change for any alternative choice $\theta_{i,j} > 0$. $\qquad\square$

We assume hyperrectangularity, which allows use to decompose the reward-maximizing policy as follows. For a given policy $\pi$, let $\boldsymbol{z}_\pi := \mathbb{E}_{s \sim \rho}[\phi(s, \pi(s))] \in \mathbb{R}^d$ denote its feature occupancy and let $\boldsymbol{z}_{\pi,j}$ be its $j$-th entry. W.l.o.g. we here assume $\boldsymbol{z}_{\pi,j} \in [-1, 1]$, but any other lower and upper bounds can be considered the same way.

We denote the optimal policy w.r.t. a reward parameter $\theta$ as $\pi^*(\theta) := \mathrm{argmax}_{\pi \in \Pi} J_\theta(\pi)$. From Assumption 2 it follows that the optimal policy $\pi^*(\theta)$ is such that $\boldsymbol{z}_{\pi^*(\theta),j} = -1$ for $\theta_j < 0$ and $\boldsymbol{z}_{\pi^*(\theta),j} = +1$ for $\theta_j > 0$. This yields an equivalence between reward parameters that have identical signs. In particular, this provides us with a class of reward parameters that induce an optimal policy w.r.t. the true reward parameter $\theta_i^*$.

**Lemma A.4.** *Let $\theta \in \mathbb{R}^d$. If $\mathrm{sign}(\theta_{i,j}^*) = \mathrm{sign}(\theta_j)$, then $\theta_{i,j}^* \cdot \boldsymbol{z}_{\pi^*(\theta),j} \geq \theta_{i,j}^* \cdot \boldsymbol{z}_{\pi^*(\tilde{\theta}),j}$ for all $\tilde{\theta} \in \mathbb{R}^d$.*

*Proof.* This follows from the structure of the optimal policies $\pi^*(\theta)$ under Assumption 2. $\qquad\square$

We fix everyone's reported parameter $\theta_{-i}$ except for individual $i$. Moreover, let $\tilde{\theta}_i \neq \theta_i^*$ and let $\tilde{\mu} := \mathrm{med}(\theta_{-i}, \tilde{\theta}_i)$ be the coordinate-wise median w.r.t. $\tilde{\theta}_i$. Similarly, let $\mu^* = \mathrm{med}(\theta_{-i}, \theta_i^*)$ be the coordinate-wise median w.r.t. $\theta^*$. We will now show that $J_{\theta_i^*}(\hat{\pi}(\mu^*)) \geq J_{\theta_i^*}(\hat{\pi}(\tilde{\mu}))$, i.e., reporting $\theta_i^*$ is the optimal strategy for individual $i$ under the Pessimistic Median of MLEs algorithm.

Since we here assume direct access to the reported parameters, given reported parameters $\theta_1, \ldots, \theta_k$, Pessimistic Median of MLEs computes the optimal policy w.r.t. the median $\mu = \mathrm{med}(\theta_1, \ldots, \theta_k)$, i.e., $\hat{\pi} = \pi^*(\mu) = \mathrm{argmax}_{\pi \in \Pi} J_\mu(\pi)$. We here assume that the $\mu$-maximizing policy is unique and otherwise use lexicographic tie-breaking. Clearly, if $\mathrm{signs}(\mu^*) = \mathrm{signs}(\tilde{\mu})$, then the policies $\hat{\pi}(\mu^*)$ and $\hat{\pi}(\tilde{\mu})$ are identical.

Next, consider any $j \in [d]$ so that $\mathrm{sign}(\mu_j^*) \neq \mathrm{sign}(\tilde{\mu}_j)$. Suppose that $\mathrm{sign}(\mu_j^*) = \mathrm{sign}(\theta_{i,j}^*)$. In this case, Lemma A.4 tells us that $\theta_{i,j}^* \cdot \boldsymbol{z}_{\hat{\pi}(\mu^*),j} \geq \theta_{i,j}^* \cdot \boldsymbol{z}_{\hat{\pi}(\tilde{\mu}),j}$. Hence, in any such dimension $j$, $\mu^*$ implies a policy that outperforms the policy maximizing $\tilde{\mu}$ w.r.t. labeler $i$'s true reward parameter $\theta_i^*$. Hence, misreporting $\tilde{\theta}_{i,j} \neq \theta_{i,j}^*$ cannot be a strictly better strategy than truthfully reporting in dimension $j$.

Suppose that $\mathrm{sign}(\mu_j^*) \neq \mathrm{sign}(\theta_{i,j}^*)$. In this case, Lemma A.3 implies that $\mathrm{sign}(\tilde{\mu}_j) = \mathrm{sign}(\mu_j^*)$, which implies $\theta_{i,j}^* \cdot \boldsymbol{z}_{\hat{\pi}(\tilde{\mu}),j} = \theta_{i,j}^* \cdot \boldsymbol{z}_{\hat{\pi}(\mu^*),j}$. Once again misreporting is never a strictly better strategy than truthfully reporting $\theta_i^*$.

We have thus confirmed that reporting $\theta_i^*$ is optimal irrespective of the other individuals' reports $\theta_{-i}$.

**Case 2 (direct access to $\theta_i$, but not $\theta_{-i}$):** Let $C_{-i}$ denote the product space of confidence sets derived from the preference data $\mathcal{D}_{-i}$ of all labelers but labeler $i$. Once again, the key lies in the observation that the assumption of hyperrectangularity implies that the labelers get to strategize over each dimension independently.

The main concern that we must alleviate is that, by taking the minimum over the confidence sets, misreporting becomes beneficial for the labelers. To this end, let $\theta_i$ be the report of individual $i$, which we for now assume to be directly observable. Given preference data $\mathcal{D}_{-i}$ and $\theta_i$, the Pessimistic Median of MLEs computes a policy maximizing

$$\min_{\theta_{-i} \in C_{-i}} \sum_{j=1}^{d} \left\langle \mathrm{med}(\theta_{-i,j}, \theta_{i,j}), \mathbb{E}_{s \sim \rho}\big[\phi(s, \pi(s))\big] \right\rangle$$

Suppose that $\theta_{i,j}^* > 0$ for $j \in [d]$. Clearly, by design of the median, for any $\theta_{-i,j}$, it follows from the same argument as in Lemma A.3 that misreporting either has no effect on the policy (if the report is $\theta_{i,j} > 0$), or can only have an adverse effect for labeler $i$ (if the report is $\theta_{i,j} < 0$). Hence, it is optimal for individual $i$ to report $\theta_i^*$ irrespective of the other individuals' reported preference data $\mathcal{D}_{-i}$ and the confidence sets that we construct.

**Case 3 (no direct access to $\theta_1, \ldots, \theta_k$):** In the previous cases, we have shown that truthfully reporting is a dominant strategy for every individual $i \in [k]$. We will now see that this is in general no longer true when an individual cannot directly share their reward parameter with the algorithm. The reason lies in unintentional changes in the sign due to estimation errors and confidence sizes.

In the following, we assume that the preference data $\mathcal{D}_{-i}$ of all labelers but labeler $i$ are fixed arbitrarily and $C_{-i}$ are the corresponding confidence sets that Pessimistic Median of MLEs constructs. From Case 2 we know that the policy

$$\hat{\pi}_i(\theta_i^*) := \underset{\pi \in \Pi}{\mathrm{argmax}} \min_{\theta_{-i} \in C_{-i}} \left\langle \mathrm{med}(\theta_i^*, \theta_{-i}), \mathbb{E}_{s \sim \rho}[\phi(s, \pi(s))] \right\rangle$$

is preferred over any other policy $\hat{\pi}(C_i)$ computed w.r.t. any confidence set $C_i$ given by

$$\hat{\pi}_i(C_i) := \underset{\pi \in \Pi}{\mathrm{argmax}} \min_{\theta_i \in C_i} \min_{\theta_{-i} \in C_{-i}} \left\langle \mathrm{med}(\theta_i, \theta_{-i}), \mathbb{E}_{s \sim \rho}[\phi(s, \pi(s))] \right\rangle,$$

i.e., $J_{\theta_i^*}(\hat{\pi}_i(\theta_i^*)) \geq J_{\theta_i^*}(\hat{\pi}_i(C_i))$ for any confidence set $C_i$.

Let us now consider the confidence set $C_i^*$ derived from $\mathcal{D}_i^*$, which is sampled according to the true reward parameter $\theta_i^*$. By construction of the confidence sets, with probability at least $1 - \delta$, it follows from Lemma A.1 that for any $\theta_i \in C_i^*$:

$$\|\theta_i^* - \theta_i\|_{\Sigma_{\mathcal{D}_i}} \leq \|\theta_i^* - \hat{\theta}_i^{\mathrm{MLE}}\|_{\Sigma_{\mathcal{D}_i}} + \|\hat{\theta}_i^{\mathrm{MLE}} - \theta_i\|_{\Sigma_{\mathcal{D}_i}} \leq 2c\sqrt{\frac{d + \log(1/\delta)}{\gamma^2 n}}.$$

We now compare the difference in return w.r.t. $\theta_i^*$ of policy $\hat{\pi}_i(\theta_i^*)$ and $\hat{\pi}_i(C_i^*)$. To do so, we decompose the difference as follows:

$$J_{\theta_i^*}(\hat{\pi}_i(\theta_i^*)) - J_{\theta_i^*}(\hat{\pi}_i(C_i^*))$$
$$= \left( J_{\theta_i^*}(\hat{\pi}_i(\theta_i^*)) - \min_{\theta_i \in C_i^*} J_{\theta_i}(\hat{\pi}_i(\theta_i^*)) \right) + \left( \min_{\theta_i \in C_i^*} J_{\theta_i}(\hat{\pi}_i(\theta_i^*)) - J_{\theta_i^*}(\hat{\pi}_i(C_i^*)) \right).$$

Using Cauchy-Schwarz, the first difference can be rewritten and bounded as

$$\max_{\theta_i \in C_i^*} \langle \theta_i^* - \theta_i, z_{\hat{\pi}(\theta_i^*)} \rangle \leq \|\theta_i^* - \theta_i\|_{\Sigma_{\mathcal{D}_i}} \|z_{\hat{\pi}(\theta_i^*)}\|_{\Sigma_{\mathcal{D}_i}^{-1}}.$$

We then further decompose the second difference into

$$\min_{\theta_i \in C_i^*} J_{\theta_i}(\hat{\pi}_i(\theta_i^*)) - J_{\theta_i^*}(\hat{\pi}_i(C_i^*))$$
$$= \left( \min_{\theta_i \in C_i^*} J_{\theta_i}(\hat{\pi}_i(\theta_i^*)) - \min_{\theta_i \in C_i^*} J_{\theta_i}(\hat{\pi}_i(C_i^*)) \right) + \left( \min_{\theta_i \in C_i^*} J_{\theta_i}(\hat{\pi}_i(C_i^*)) - J_{\theta_i^*}(\hat{\pi}_i(C_i^*)) \right).$$

By definition of $\hat{\pi}_i(C_i^*)$, we have $\min_{\theta_i \in C_i^*}\langle\theta_i, z_{\hat{\pi}_i(C_i^*)}\rangle \geq \min_{\theta_i \in C_i^*}\langle\theta_i, z_{\hat{\pi}(\theta_i^*)}\rangle$ so that the first expression on the right hand side is less or equal to zero. Since $\theta_i^* \in C_i^*$, we also know that $\min_{\theta_i \in C_i^*} J_{\theta_i}(\pi) \leq J_{\theta_i^*}(\pi)$ for all $\pi \in \Pi$. Thus, on the good event when $\theta_i^* \in C_i^*$, we obtain:

$$J_{\theta_i^*}(\hat{\pi}_i(\theta_i^*)) - J_{\theta_i^*}(\hat{\pi}_i(C_i^*)) \leq c\sqrt{\frac{d + \log(1/\delta)}{\gamma^2 n}} \cdot \|\mathbb{E}_{s\sim\rho}[\phi(s, \hat{\pi}_i(\theta_i^*)(s))]\|_{\Sigma_{\mathcal{D}_i}^{-1}}$$

$$\leq c\sqrt{\frac{d + \log(1/\delta)}{\gamma^2 n}} \cdot \max_{\pi\in\Pi}\|\mathbb{E}_{s\sim\rho}[\phi(s, \pi(s))]\|_{\Sigma_{\mathcal{D}_i}^{-1}}.$$

Note that the coverage coefficient on the right can be written as $\|\Sigma_{\mathcal{D}_i}^{-1/2}\mathbb{E}_{s\sim\rho}[\phi(s, \pi(s))]\|_2$.

We have here (arguably coarsely) upper bounded the coverage of $\hat{\pi}(\theta_i^*)$ by the uniform policy coverage $\kappa_i := \max_\pi\|\mathbb{E}_{s\sim\rho}[\phi(s, \pi(s))]\|_{\Sigma_{\mathcal{D}_i}^{-1}}$ of labeler's $i$ data. We must do this here as the policy $\hat{\pi}_i(\theta_i^*)$ notably depends on the other labeler's reported preferences $\mathcal{D}_{-i}$ and is thus hard to control or express explicitly. Overall, we have thus shown that being truthful is an approximately dominant strategy for labeler $i$ under Pessimistic Median of MLEs. Hence, Pessimistic MoMLEs is $\mathcal{O}(\kappa_i\sqrt{d/n})$-strategyproof.

**Remark A.5.** *As we wish to ensure strategyproofness, i.e., truthfulness is a dominant strategy, we could not control the needed coverage carefully, but had to take a worst-case perspective and consider uniform coverage of all policies as quantified by $\kappa_i$. Naturally, we would expect to improve upon this when considering incentive-compatibility instead of strategyproofness, i.e., showing that truthfulness forms an equilibrium but is not necessarily a dominant strategy profile. In that case, one can show that Pessimistic Median of MLEs is approximately incentive-compatible where instead of the uniform policy coverage the coverage of the output $\hat{\pi}^*$ of Pessimistic Median of MLEs given that everyone reports truthfully is enough. In other words, the coverage coefficient is given by $\|\mathbb{E}_{s\sim\rho}[\phi(s, \hat{\pi}^*(s))]\|_{\Sigma_{\mathcal{D}_i}^{-1}} \leq \kappa_i$.*

$\square$

### A.6  Proof of Theorem 4.2

**Theorem 4.2.** *Let $\hat{\pi}$ be the output of the Pessimistic Median of MLEs algorithm and suppose that all labelers report truthfully. With probability at least $1 - \delta$:*

$$\text{SubOpt}(\hat{\pi}) \leq const \cdot \left(\sqrt{\frac{d\log(k/\delta)}{k}} + \max_{i\in[k]}\kappa_i^* \cdot k\sqrt{\frac{d + \log(k/\delta)}{n}}\right) \tag{2}$$

*where $\kappa_i^* := \|\mathbb{E}_{s\sim\rho}[\phi(s, \pi^*(s))]\|_{\Sigma_{\mathcal{D}_i}^{-1}}$ is the optimal policy coverage of labeler $i$.*

*Proof.* We will decompose the suboptimality in various ways. To this end, let $\pi^*$ denote the policy that maximizes social welfare and let $\hat{\pi}$ denote the policy computed by Pessimistic Median of MLEs. Recall the definition of the set of medians w.r.t. confidence sets $C_1, \ldots, C_k$ as $\mathscr{C} := \{\text{med}(\theta_1, \ldots, \theta_k) : \theta_i \in C_i\}$ and let $\mathscr{A} := \{\frac{1}{k}\sum_{i=1}^k \theta_i : \theta_i \in C_i\}$ denote the set of averages. For convenience, we define for any $\pi$:

$$z_\pi := \mathbb{E}_{s\sim\rho}[\phi(s, \pi(s))].$$

Moreover, we let

$$\theta_{\text{avg}}^* := \text{avg}(\theta_1^*, \ldots, \theta_k^*) \quad \text{and} \quad \theta_{\text{med}}^* := \text{med}(\theta_1^*, \ldots, \theta_k^*)$$

correspond to the true average and median, respectively. We now decompose the suboptimality as follows:

$$\text{SubOpt}(\hat{\pi}) = \frac{1}{k}\sum_{i=1}^k \langle\theta_i^*, z_{\pi^*}\rangle - \langle\theta_i^*, z_{\hat{\pi}}\rangle$$

$$= \langle\theta_{\text{avg}}^*, z_{\pi^*}\rangle - \langle\theta_{\text{avg}}^*, z_{\hat{\pi}}\rangle$$

$$= \underbrace{\left(\langle\theta_{\text{avg}}^*, z_{\pi^*}\rangle - \min_{\theta\in\mathscr{A}}\langle\theta, z_{\pi^*}\rangle\right)}_{(I)} + \underbrace{\left(\min_{\theta\in\mathscr{A}}\langle\theta, z_{\pi^*}\rangle - \langle\theta_{\text{avg}}^*, z_{\hat{\pi}}\rangle\right)}_{(II)}$$

In the following, we work on the good event such that $\theta_i^* \in C_i$ for all $i \in [k]$. Using a union bound, we can show that this event occurs with probability at least $1 - \frac{k}{d}$.

We can bound the first term *(I)* using that the confidences concentrate around the true parameter at a rate of $\sqrt{d/n}$ according to Lemma A.1 and considering the worst-case coverage of the optimal policy over all labeler's data. For some constant $c > 0$, this yields

$$
\langle \theta_{\text{avg}}^*, z_{\pi^*} \rangle - \min_{\theta \in \mathscr{A}} \langle \theta, z_{\pi^*} \rangle = \max_{\theta \in \mathscr{A}} \langle \theta_{\text{avg}}^* - \theta, z_{\pi^*} \rangle
$$

$$
= \frac{1}{k} \max_{\theta_1 \in C_1} \ldots \max_{\theta_k \in C_k} \sum_{i=1}^k \langle \theta_i^* - \theta_i, z_{\pi^*} \rangle
$$

$$
= \frac{1}{k} \sum_{i=1}^k \max_{\theta_i \in C_i} \langle \theta_i^* - \theta_i, z_{\pi^*} \rangle
$$

$$
\leq \frac{1}{k} \sum_{i=1}^k \max_{\theta_i \in C_i} \| \theta_i^* - \theta_i \|_{\Sigma_{\mathcal{D}_i}} \| z_{\pi^*} \|_{\Sigma_{\mathcal{D}_i}^{-1}}
$$

$$
\leq c \sqrt{\frac{d + \log(1/\delta)}{\gamma^2 n}} \cdot \frac{1}{k} \sum_{i=1}^k \| z_{\pi^*} \|_{\Sigma_{\mathcal{D}_i}^{-1}}
$$

$$
\leq c \sqrt{\frac{d + \log(1/\delta)}{\gamma^2 n}} \cdot \max_{i \in [k]} \| z_{\pi^*} \|_{\Sigma_{\mathcal{D}_i}^{-1}}.
$$

Bounding the second term *(II)* is more involved as the policy $\hat{\pi}$ is not maximizing the average but the pessimistic median. We further decompose the second term into four parts as follows:

$$
\min_{\theta \in \mathscr{A}} \langle \theta, z_{\pi^*} \rangle - \langle \theta_{\text{avg}}^*, z_{\hat{\pi}} \rangle = \left( \min_{\theta \in \mathscr{A}} \langle \theta, z_{\pi^*} \rangle - \min_{\theta \in \mathscr{C}} \langle \theta, z_{\pi^*} \rangle \right) + \left( \min_{\theta \in \mathscr{C}} \langle \theta, z_{\pi^*} \rangle - \min_{\theta \in \mathscr{C}} \langle \theta, z_{\hat{\pi}} \rangle \right)
$$

$$
+ \left( \min_{\theta \in \mathscr{C}} \langle \theta, z_{\hat{\pi}} \rangle - \langle \theta_{\text{med}}^*, z_{\hat{\pi}} \rangle \right) + \left( \langle \theta_{\text{med}}^*, z_{\hat{\pi}} \rangle - \langle \theta_{\text{avg}}^*, z_{\hat{\pi}} \rangle \right).
$$

We first show that the second and third term are less or equal to zero. We have

$$
\min_{\theta \in \mathscr{C}} \langle \theta, z_{\pi^*} \rangle \leq \min_{\theta \in \mathscr{C}} \langle \theta, z_{\hat{\pi}} \rangle,
$$

since $\hat{\pi}$ maximizes $\min_{\theta \in \mathscr{C}} \langle \theta, z_{\pi} \rangle$ by definition of the Pessimistic Median of MLEs. Moreover, we see that

$$
\min_{\theta \in \mathscr{C}} \langle \theta, z_{\hat{\pi}} \rangle \leq \langle \theta_{\text{med}}^*, z_{\hat{\pi}} \rangle,
$$

as the true median is contained in the confidence set $\mathscr{C}$ on the good event when $\theta_i^* \in C_i$. Hence, both the second and third term can be bounded from above by zero.

To bound the first term, we once again decompose the expression as follows:

$$
\min_{\theta \in \mathscr{A}} \langle \theta, z_{\pi^*} \rangle - \min_{\theta \in \mathscr{C}} \langle \theta, z_{\pi^*} \rangle = \underbrace{\min_{\theta \in \mathscr{A}} \langle \theta - \theta_{\text{avg}}^*, z_{\pi^*} \rangle}_{(a)} + \underbrace{\langle \theta_{\text{avg}}^* - \theta_{\text{med}}^*, z_{\pi^*} \rangle}_{(b)} + \underbrace{\max_{\theta \in \mathscr{C}} \langle \theta_{\text{med}}^* - \theta, z_{\pi^*} \rangle}_{(c)}.
$$

$$
\tag{3}
$$

Similarly to before, using Lemma A.1, we bound $(a)$ as

$$
\min_{\theta \in \mathscr{A}} \langle \theta - \theta_{\text{avg}}^*, z_{\pi^*} \rangle \leq c \sqrt{\frac{d + \log(1/\delta)}{\gamma^2 n}} \cdot \max_{i \in [k]} \| z_{\pi^*} \|_{\Sigma_{\mathcal{D}_i}^{-1}}.
$$

For $(b)$, it follows from Cauchy-Schwarz and Lemma A.2 that

$$
\langle \theta_{\text{avg}}^* - \theta_{\text{med}}^*, z_{\pi^*} \rangle \leq \| \theta_{\text{avg}}^* - \theta_{\text{med}}^* \|_2 \| z_{\pi^*} \|_2 \leq c \sqrt{\frac{d \log(1/\delta)}{k}} \cdot \| z_{\pi^*} \|_2.
$$

$$
\tag{4}
$$

For $(c)$, first note that we can write the difference between two medians as the telescoping sum

$$\text{med}(\theta_1^*, \ldots, \theta_k^*) - \text{med}(\theta_1, \ldots, \theta_k)$$
$$= \sum_{i=1}^{k} \text{med}(\theta_1^*, \ldots, \theta_i^*, \theta_{i+1}, \ldots, \theta_k) - \text{med}(\theta_1^*, \ldots, \theta_{i-1}^*, \theta_i, \ldots, \theta_k).$$

By definition of the median, each difference on the right hand side can be bounded in terms of the difference $\theta_i^* - \theta_i$. Using Lemma A.1 and the fact that $\theta_i \in C_i$ for all $i \in [k]$, we obtain

$$\max_{\theta \in \mathscr{C}} \langle \theta_{\text{med}}^* - \theta, \boldsymbol{z}_{\pi^*} \rangle \leq \sum_{i=1}^{k} \|\theta_i^* - \theta_i\|_{\Sigma_{\mathcal{D}_i}} \|\boldsymbol{z}_{\pi^*}\|_{\Sigma_{\mathcal{D}_i}^{-1}}$$
$$\leq ck \sqrt{\frac{d + \log(1/\delta)}{\gamma^2 n}} \cdot \|\boldsymbol{z}_{\pi^*}\|_{\Sigma_{\mathcal{D}_i}^{-1}}.$$

The proof is complete by combining these bounds. $\qquad\square$

### A.7 Proof of Proposition 4.3

**Proposition 4.3.** *When the labelers report their preferences according to any weakly dominant strategy under Pessimistic Median of MLEs, with probability at least $1 - \delta$, the output $\hat{\pi}$ satisfies:*

$$\text{SubOpt}(\hat{\pi}) \leq const \cdot \left( \sqrt{\frac{d \log(k/\delta)}{k}} + \max_{i \in [k]} \kappa_i^* \cdot k \sqrt{\frac{d + \log(k/\delta)}{n}} \right).$$

*where $\kappa_i^* := \|\mathbb{E}_{s \sim \rho}[\phi(s, \pi^*(s))]\|_{\Sigma_{\mathcal{D}_i}^{-1}}$ is the optimal policy coverage of labeler $i$.*

*Proof.* Let $\theta_i \in \mathbb{R}^d$ be the reward parameter according to which labeler $i \in [k]$ samples its preferences under a weakly dominant strategy. The intuition for the result is fairly straightforward so that we describe it here first. First of all, we have seen in the proof of Theorem 4.1 that due to the median rule a labeler cannot achieve an individually better outcome by misreporting the sign of its reward parameter (see Lemma A.3 and Lemma A.4). As a result, $\theta_i$ will have identical signs to $\theta_i^*$ but potentially exaggerate its magnitude. Crucially, such exaggeration cannot worsen the suboptimality as it only helps to prevent flipped signs as we are taking the worst-case over confidence sets. Here, it is also worth noting that the primary reasons why Pessimistic Median of MLEs is approximately strategyproof are the estimation errors and the pessimism selection of the median over potentially large confidence sets.

**Any weakly dominant strategy must preserve signs.** Assume for contradiction that there exists some coordinate $j$ such that $\theta_{i,j}^* > 0$ but the labeler's chosen reward parameter is such that $\theta_{i,j} < 0$ (or similarly $\theta_{i,j}^* < 0$ but $\theta_{i,j} > 0$). By the hyperrectangular assumption, the Pessimistic Median of MLEs algorithm outputs a policy that maximizes each dimension of the feature space independently. Specifically, $\boldsymbol{z}_{\hat{\pi},j} = \mathbb{E}_{s \sim \rho}[\phi(s, \hat{\pi}(s))]_j$ will be positive if the considered median is positive and vice versa. Hence, by nature of the median, flipping the sign of coordinate $j$ can only have an adverse effect for labeler $i$ (see Lemma A.3) and no such strategy can be weakly dominant.

By the same argument, if $\theta_{i,j}^* < 0$ but $\theta_{i,j} > 0$, then labeler $i$ would risk pushing the aggregator's dimension $j$ to be positive, contrary to its true negative preference, and thus risk reducing its true utility in that dimension. Hence it cannot be a weakly dominant strategy to flip signs in that scenario either. Consequently, in every dimension $j$, a weakly dominant report $\theta_{i,j}$ must preserve $\text{sign}(\theta_{i,j}) = \text{sign}(\theta_{i,j}^*)$.

**Exaggeration benefits Pessimistic Median of MLEs.** By the hyperrectangular ("sign-based") structure, the decision in each coordinate $j$ of the learned policy depends essentially on whether the aggregated median is positive or negative. Pessimistic Median of MLEs aggregates each labeler $i$'s confidence set $C_i$ by taking a coordinate-wise median over a selection $\theta_i \in C_i$. Thus, to form the median, it chooses exactly one $\theta_{i,j}$ from each $C_i$ and then takes the median value among these $k$ numbers. Assume that the labeler $i$'s original (w.l.o.g.) positive coordinate is $\theta_{i,j}^*$, whereas its inflated coordinate is $\theta_{i,j} > \theta_{i,j}^*$. Under the inflated reported reward parameter, the labeler's MLE and confidence set for dimension $j$ shift toward strictly larger positive values (note that the covariance

matrix $\Sigma_{\mathcal{D}_i}$ is positive definite). Consequently, the set of considered medians $\mu_j$ for $\mu \in \mathscr{C}$ (i.e. all possible ways to pick $\theta_{i,j} \in C_i$ for $i = 1, \ldots, k$ and take their coordinate-wise median) does not move down: it can only stay the same or shift to more positive values. Intuitively, replacing one of the $i$ entries by a strictly larger positive number cannot decrease the median.

Hence, when labeler $i$ is misreporting $\theta_{i,j}$ such that $\theta_{i,j} > \theta_{i,j}^*$ (while keeping the same sign), this cannot worsen the suboptimality of the final policy, but only, in some special cases, strictly lower suboptimality by "protecting" the sign within the confidence set. Since this argument holds for any dimension $j$, it follows that an entire sign-preserving inflation by labeler $i$ cannot yield a higher suboptimality than the truthful report would. $\qquad\square$

## A.8 Proof of Corollary 4.5

**Corollary 4.5.** *When there is only a single labeler, with probability at least $1 - \delta$:*

$$\mathrm{SubOpt}(\hat{\pi}) \leq const \cdot \kappa_1^* \sqrt{\frac{d + \log(k/\delta)}{n}}.$$

*When all $k$ labelers have the same reward function, with probability at least $1 - \delta$:*

$$\mathrm{SubOpt}(\hat{\pi}) \leq const \cdot \max_{i \in [k]} \kappa_i^* \cdot k \sqrt{\frac{d + \log(k/\delta)}{n}}.$$

*Proof.* When $k = 1$ the claimed result follows directly from setting $k = 1$ in our previous suboptimality bounds (see Theorem 4.2).

Next, suppose that all $k \geq 1$ labelers have the same reward parameter $\theta^* = \theta_1^* = \cdots = \theta_k^*$. As a result, the true average and median coincide and we have $\theta^* = \theta_{\mathrm{avg}}^* = \theta_{\mathrm{med}}^*$. To bound the suboptimality of the Pessimistic Median of MLEs algorithm in this special case we take the same steps as in the proof of Theorem 4.2 in Section A.6 with the difference that the expression $(b)$ in equation (3) is zero since $\theta^* = \theta_{\mathrm{avg}}^* = \theta_{\mathrm{med}}^*$. This yields the claimed upper bound. $\qquad\square$

## A.9 Proof of Corollary 4.6

**Corollary 4.6.** *Suppose $\mathcal{W}(\pi^*) > 0$ is constant. When the number of samples is sufficiently large and provide sufficient coverage of the optimal policy, with probability at least $1 - \delta$, the approximation ratio of the Pessimistic Median of MLEs algorithm is given by $\alpha(\rho, \hat{\pi}) \geq 1 - \mathcal{O}\big(\sqrt{d \log(k/\delta)/k}\big)$.*

*Proof.* For $n$ sufficiently large and sufficient coverage of the optimal policy, Theorem 4.2 implies that with probability at least $1 - \delta$:

$$\mathrm{SubOpt}(\hat{\pi}) \coloneqq \mathcal{W}(\pi^*) - \mathcal{W}(\hat{\pi}) \leq c \sqrt{\frac{d \log(k/\delta)}{n}}$$

for some constant $c > 0$. As a result, the approximation ratio is upper bounded as

$$\alpha(\rho, \hat{\pi}) \coloneqq \frac{\mathcal{W}(\hat{\pi})}{\mathcal{W}(\pi^*)} = 1 - \frac{\mathcal{W}(\pi^*) - \mathcal{W}(\hat{\pi})}{\mathcal{W}(\pi^*)} \geq 1 - c \sqrt{\frac{d \log(k/\delta)}{n}},$$

where we used that $\mathcal{W}(\pi^*) > 0$ is constant by assumption.

$\qquad\square$

## A.10 Proof of Theorem 5.1 and Theorem 5.2

*Proof.* We can prove Theorem 5.1 and Theorem 5.2 in a similar way we proved the analogous results in the contextual bandit problem. We refrain from reiterating and restating all necessary steps to prove these results as they are almost identical to before. Most importantly, a similar MLE concentration bound holds for MDPs as for contextual bandits.

**Lemma A.6** (MLE Concentration Bound for MDPs). *With probability at least $1 - \delta$,*

$$\|\hat{\theta}_i^{\text{MLE}} - \theta_i^*\|_{\Sigma_{\mathcal{D}_i}} \leq const \cdot \sqrt{\frac{d + \log(1/\delta)}{\gamma^2 n}},$$

*where $\gamma := 1/(2 + \exp(-HLB) + \exp(HLB))$. The covariance matrix $\Sigma_{\mathcal{D}_i}$ is given by $\Sigma_{\mathcal{D}_i} = \sum_{j=1}^n x^{i,j}(x^{i,j})^\top$ where $x^{i,j} = \sum_{h=1}^H (\phi(s_h^{i,j}, a_h^{i,j}) - \phi(\bar{s}_h^{i,j}, \bar{a}_h^{i,j}))$ with $s_1^{i,j} = \bar{s}_1^{i,j} = s^{i,j}$.*

Swapping the initial state distribution $\rho$ (i.e., context distribution) for the the state occupancy $q_\pi$ as defined in Section 5, we can follow the same line of argument as in Section A.5 to prove Theorem 5.1. $\qquad\square$

# B  Computational Complexity

We now consider the computational complexity of computing the pessimistic median return. First, we consider the contextual bandits formulation, and then consider the general MDP setting.

## B.1  Contextual Bandits

Recall that we construct the confidence sets

$$C_i = \{\theta \in \mathbb{R}^d : \|\hat{\theta}_i^{\text{MLE}} - \theta\|_{\Sigma_{\mathcal{D}_i}} \leq f(d, n, \delta)\}.$$

Then, the (coordinate-wise) median confidence set is defined as

$$\mathscr{C} = \{\text{med}(\theta_1, \ldots, \theta_k) : \theta_i \in C_i \; \forall i \in [k]\},$$

and we aim to solve the following optimization problem:

$$\max_{\pi \in \Pi} \underline{\mathcal{W}}(\pi) := \min_{\theta \in \mathscr{C}} \mathbb{E}_{s \sim \rho}\left[\langle \theta, \phi(s, \pi(s)) \rangle\right]$$

In a first step, we show that the function $\underline{\mathcal{W}}(\pi)$ is concave. Indeed, consider two policies $\pi_1$, and $\pi_2$. Then,

$$\underline{\mathcal{W}}(\alpha\pi_1 + (1 - \alpha)\pi_2) = \min_{\theta \in \mathscr{C}} \mathbb{E}_{s \sim \rho}\left[\sum_a (\alpha\pi_1(a) + (1 - \alpha)\pi_2(a))\langle \theta, \phi(s, a)\rangle\right]$$

$$\geq \min_{\theta \in \mathscr{C}} \mathbb{E}_{s \sim \rho}\left[\sum_a \alpha\pi_1(a)\langle \theta, \phi(s, a)\rangle\right] + \min_{\theta \in \mathscr{C}} \mathbb{E}_{s \sim \rho}\left[\sum_a (1 - \alpha)\pi_2(a)\langle \theta, \phi(s, a)\rangle\right]$$

$$= \alpha \cdot \underline{\mathcal{W}}(\pi_1) + (1 - \alpha) \cdot \underline{\mathcal{W}}(\pi_2).$$

Therefore, $\underline{\mathcal{W}}(\cdot)$ can be efficiently optimized using projected gradient ascent as long as we can compute the gradient efficiently. For a given $\pi$, we have $\nabla_\pi \underline{\mathcal{W}}(\pi)_{(s,a)} = \rho(s)\langle\phi(s, a), \theta^\star\rangle$ where

$$\theta^\star \in \operatorname*{argmin}_{\theta \in \mathscr{C}} \mathbb{E}_{s \sim \rho}\left[\langle\theta, \phi(s, \pi(s))\rangle\right]. \tag{5}$$

In order to show that the gradient $\nabla_\pi \underline{\mathcal{W}}(\pi)$ can be efficiently computed, we need to show that $\theta^\star$ can be efficiently computed.

The set $\mathscr{C}$ can be arbitrary, but we can write down the following equivalent optimization problem involving linear and quadratic constraints:

$$\min_{\theta, \{\theta_i\}_{i \in [k]}} \mathbb{E}_{s \sim \rho}\left[\langle\theta, \phi(s, \pi(s))\rangle\right]$$

$$\text{s.t. } \|\hat{\theta}_i^{\text{MLE}} - \theta_i\|_{\Sigma_{\mathcal{D}_i}} \leq f(d, n, \delta) \; \forall i \in [k] \tag{6}$$

$$\sum_{i=1}^k |\theta(j) - \theta_i(j)| \leq \sum_{i=1}^k |\theta_\ell(j) - \theta_i(j)| \; \forall \ell \in [k], \forall j \in [d]$$

The first set of constraints encode that $\theta_i \in C_i$ for each $i \in [k]$. The second set of constraints encode that $\theta(j)$ is the median of $\theta_1(j), \ldots, \theta_k(j)$ for each coordinate $j \in [d]$. The above optimization

problem might be non-convex, and instead we will consider the following alternate optimization problem:

$$\min_{\theta,\{\theta_i\}_{i\in[k]},z} \mathbb{E}_{s\sim\rho}[\langle\theta,\phi(s,\pi(s))\rangle] + M\sum_{i,j} z_{i,j}$$

$$\text{s.t. } \theta_i \in C_i \ \forall i \in [k] \tag{7}$$

$$z_{i,j} \geq \theta(j) - \theta_i(j), \ z_{i,j} \geq \theta_i(j) - \theta(j) \ \forall i \in [k], j \in [d]$$

The next lemma shows that we can choose $M$ and $n$ to recover an approximate solution of the original optimization problem (6).

**Lemma B.1.** *Suppose* $(\theta^1, \{\theta_i^1\}_{i\in[k]})$ *is an optimal solution to the optimization problem* (6)*, and* $(\theta^2, \{\theta_i^2\}_{i\in[k]}, z^2)$ *is an optimal solution to the optimization problem* (7)*. Then,*

$$\mathbb{E}_{s\sim\rho}[\langle\theta^2,\phi(s,\pi(s))\rangle] \leq \mathbb{E}_{s\sim\rho}[\langle\theta^1,\phi(s,\pi(s))\rangle] + M\sum_i \frac{2\sqrt{d}}{\lambda_{\min}(\Sigma_{\mathcal{D}_i})} f(d,n,\delta)$$

*and*

$$\sum_{i,j}|\theta^2(j) - \theta_i^2(j)| - \sum_{i,j}\left|\tilde{\theta}(j) - \theta_i^2(j)\right| \leq \frac{2BL}{M} \ \forall\tilde{\theta}.$$

*Proof.* Let us define $z_{i,j}^1 = |\theta^1(j) - \theta_i^1(j)|$. Then we have the following inequality:

$$\mathbb{E}_{s\sim\rho}[\langle\theta^2,\phi(s,\pi(s))\rangle] + M\sum_{i,j} z_{i,j}^2 \leq \mathbb{E}_{s\sim\rho}[\langle\theta^1,\phi(s,\pi(s))\rangle] + M\sum_{i,j} z_{i,j}^1.$$

Without loss of generality, we can assume $z_{i,j}^2 = |\theta^2(j) - \theta_i^2(j)|$. Hence,

$$z_{i,j}^2 = |\theta^2(j) - \theta_i^2(j)| = |\theta^2(j) - \theta_i^1(j) + \theta_i^1(j) - \theta_i^2(j)| \geq |\theta^2(j) - \theta_i^1(j)| - |\theta_i^1(j) - \theta_i^2(j)|.$$

Substituting this bound, we obtain

$$\mathbb{E}_{s\sim\rho}[\langle\theta^2,\phi(s,\pi(s))\rangle] - \mathbb{E}_{s\sim\rho}[\langle\theta^1,\phi(s,\pi(s))\rangle]$$

$$\leq -M\sum_{i,j}|\theta^2(j) - \theta_i^1(j)| + M\sum_{i,j}|\theta^1(j) - \theta_i^1(j)| + M\sum_{i,j}|\theta_i^1(j) - \theta_i^2(j)|$$

$$\leq M\sum_i \left\|\theta_i^1 - \theta_i^2\right\|_1$$

$$\leq M\sum_i \sqrt{d}\left\|\theta_i^1 - \theta_i^2\right\|_2$$

$$\leq M\sum_i \frac{\sqrt{d}}{\lambda_{\min}(\Sigma_{\mathcal{D}_i})}\left\|\theta_i^1 - \theta_i^2\right\|_{\Sigma_{\mathcal{D}_i}} \leq M\sum_i \frac{2\sqrt{d}}{\lambda_{\min}(\Sigma_{\mathcal{D}_i})} f(d,n,\delta).$$

The second inequality follows since $\theta^1$ is a coordinate-wise median of the parameters $\{\theta_i^1\}_{i\in[k]}$. We also use the assumption that under uniform coverage $\lambda_{\min}(\Sigma_{\mathcal{D}_i}) > 0$.

We now bound the violation of constraints of the solution $\theta^2$. Indeed for any $\tilde{\theta}, \{\theta_i^2\}_{i\in[k]}$ and $\left\{|\tilde{\theta}(j) - \theta_i^2(j)|\right\}_{i,j}$, since $\theta^2$ as $\theta^2, \{\theta_i^2\}_{i\in[k]}, \{z_{i,j}\}_{i,j}$ is feasible solution to the optimization problem, we have (7):

$$\mathbb{E}_{s\sim\rho}[\langle\theta^2,\phi(s,\pi(s))\rangle] + M\sum_{i,j} z_{i,j}^2 \leq \mathbb{E}_{s\sim\rho}[\langle\tilde{\theta},\phi(s,\pi(s))\rangle] + M\sum_{i,j}\left|\tilde{\theta}(j) - \theta_i^2(j)\right|.$$

After rearranging this yields

$$\sum_{i,j}|\theta^2(j) - \theta_i^2(j)| - \sum_{i,j}\left|\tilde{\theta}(j) - \theta_i^2(j)\right| \leq \frac{1}{M}\mathbb{E}_{s\sim\rho}[\langle\tilde{\theta} - \theta^2,\phi(s,\pi(s))\rangle] \leq \frac{2BL}{M}.$$

$\square$

Since $f(d, n, \delta) = O\left(\sqrt{\frac{d + \log(k/\delta)}{n}}\right)$ we can choose $M = \frac{2BL}{\varepsilon}$ and $n \geq \frac{4B^2 L^2 k^2 (d + \log(k/\delta))}{(\min_i \lambda_{\min}(\Sigma_{\mathcal{D}_i})^2} \cdot \frac{1}{\varepsilon^4}$ and observe that $\theta^2$ is $\varepsilon$-approximately optimal and $\varepsilon$-approximate coordinate-wise median of the parameters $\{\theta_i^2\}_{i \in [k]}$.

**General setting.** Next, we consider the general setting where the state space can be arbitrarily large. We will assume that the policies are parametrized by a class of parameters $\Psi \subseteq \mathbb{R}^d$, i.e., $\Pi = \{\pi_\psi : \psi \in \Psi\}$. For example, the softmax parametrization models $\Pi$ as the following class of policies:

$$\Pi = \left\{ \pi_\psi(a \mid s) = \frac{\exp(\psi^\top \phi(s, a))}{\sum_b \exp(\psi^\top \phi(s, b))} \; \forall s, a : \|\psi\|_2 \leq B \right\}.$$

We now aim to solve the following optimization problem:

$$\max_{\psi \in \Psi} \underline{\mathcal{W}}(\psi) := \min_{\theta \in \mathscr{C}} \mathbb{E}_{s \sim \rho} \left[ \sum_a \pi_\psi(a|s) \langle \theta, \phi(s, a) \rangle \right].$$

The gradient of the objective is given by

$$\nabla_\psi \underline{\mathcal{W}}(\psi) = \mathbb{E}_{s \sim \rho} \left[ \sum_a \nabla_\psi \pi_\psi(a|s) \langle \theta^\star, \phi(s, a) \rangle \right]$$

where

$$\theta^\star \in \operatorname*{argmin}_{\theta \in \mathscr{C}} \mathbb{E}_{s \sim \rho} \left[ \langle \theta, \phi(s, \pi_\psi(s)) \rangle \right].$$

The above optimization is finite-dimensional ($O(dk)$) even when the number of states is very large and can be approximated using optimization problem (7). Thus, we can perform projected gradient ascent steps to solve the optimization problem $\max_{\psi \in \Psi} \underline{\mathcal{W}}(\psi)$. However, unlike the tabular setting, the objective is no longer concave. It is known that under softmax parametrization, the expected return satisfies a non-uniform Polyak-Lojasiewicz (PL) condition [25] which guarantees linear convergence of gradient ascent method. We believe that similar conditions should hold for the function $\underline{\mathcal{W}}(\psi)$ but leave the verification to the future.

## B.2 Extension to Markov Decision Processes

We start with the assumption of a tabular MDP. We aim to solve the following optimization problem:

$$\max_{\pi \in \Pi} \underline{\mathcal{W}}(\pi) := \min_{\theta \in \mathscr{C}} \mathbb{E}_{s \sim q_\pi} [\langle \theta, \phi(s, \pi(s)) \rangle].$$

$\underline{\mathcal{W}}(\pi)$ is a non-convex function of policy $\pi$. However, it is a concave function of $q_\pi$, the state-action occupancy measure of policy $\pi$. We can write down the above optimization problem in terms of state-action occupancy measure as follows:

$$\max_q \min_{\theta \in \mathscr{C}} \frac{1}{H} \sum_{h=1}^H \sum_{s,a} q_h(s, a) \langle \theta, \phi(s, a) \rangle$$

$$\text{s.t.} \sum_a q_1(s, a) = \rho(s) \; \forall s$$

$$\sum_b q_{h+1}(s, b) = \sum_{s',a} q_h(s', a) P(s|s', a) \; \forall s \; \forall h \in [H-1].$$

Here, the last two constraints encode Bellman-flow conditions which ensure that the solution $q$ is a valid state-action occupancy measure. Now we can write down stochastic gradient ascent step as $q_{t+1} = \mathrm{Proj}(q_t + \eta \nabla \underline{\mathcal{W}}(q_t))$. Here the $\mathrm{Proj}(\cdot)$ refers to projection onto the feasible set defined by the flow conditions. As the number of states and actions are finite and small, the projection step can be computed efficiently. We now verify that the gradient $\nabla \underline{W}(q)$ can be computed efficiently. The gradient is given by

$$\nabla \underline{\mathcal{W}}(q) = \frac{1}{H} \sum_{h=1}^H \sum_{s,a} q_h(s, a) \langle \theta^\star, \phi(s, a) \rangle$$

where

$$\theta^\star \in \operatorname*{argmin}_{\theta \in \mathscr{C}} \; \frac{1}{H} \sum_{h=1}^{H} \sum_{s,a} q_h(s,a) \langle \theta, \phi(s,a) \rangle.$$

Now we can proceed similarly to the contextual bandits setting and compute an approximate solution of the optimization problem above.

$$\min_{\theta, \{\theta_i\}_{i \in [k]}, z} \frac{1}{H} \sum_{h=1}^{H} \sum_{s,a} q_h(s,a) \langle \theta, \phi(s,a) \rangle + M \sum_{i,j} z_{i,j} \tag{8}$$
$$\text{s.t. } \theta_i \in C_i \; \forall i \in [k]$$
$$z_{i,j} \geq \theta(j) - \theta_i(j), \; z_{i,j} \geq \theta_i(j) - \theta(j) \; \forall i \in [k], j \in [d]$$

**General Setting**: In the general setting, computing the projection step becomes infeasible as the number of states (and constraints) can be very large and possibly infinite. Instead, we again adopt a policy parametrization as discussed in the previous subsection. In particular, we assume that the policies are parametrized by a class of parameters $\Psi \subseteq \mathbb{R}^d$, i.e., $\Pi = \{\pi_\psi : \psi \in \Psi\}$. We now aim to solve the following optimization problem:

$$\max_{\psi \in \Psi} \underline{\mathcal{W}}(\psi) := \min_{\theta \in \mathscr{C}} \mathbb{E}_{s \sim \rho} \left[ V_\theta^{\pi_\psi}(s) \right].$$

The gradient of the objective is given by

$$\nabla_\psi \underline{\mathcal{W}}(\psi) = \mathbb{E}_{s \sim \rho} \left[ \nabla_\psi V_{\theta^\star}^{\pi_\psi}(s) \right]$$

where

$$\theta^\star \in \operatorname*{argmin}_{\theta \in \mathscr{C}} \mathbb{E}_{(s,a) \sim q_{\pi_\psi}} \left[ \langle \theta, \phi(s,a) \rangle \right].$$

## C  Experiments: Simulating Strategic Preference Labeling

We here conduct small-scale synthetic experiments that simulate strategic preference learning and serve as a preliminary empirical evaluation of the proposed methodology.

**Experimental Setup.**   We simulate strategic labeling behavior by performing approximate gradient ascent (i.e., simultaneous perturbation stochastic approximation) on each labeler's utility $J_i(\hat{\pi})$ w.r.t. the labelers' internal reward parameters $\hat{\theta}_i$, which govern their preference distribution $\mathbb{P}_{\hat{\theta}_i}$. We adopt this simulation approach from prior work on strategic contextual bandits [23]. Each labeler is initialized at their ground-truth reward vector $\theta_i^*$, which is sampled from a multivariate Gaussian. Labeler strategies are optimized for 200 steps. Since this process requires repeatedly re-labeling comparisons and re-running each algorithm, we focus on small problem settings in a contextual bandit formulation. All results are averaged over 5 random seeds, and we report standard errors. The results below are for embedding dimension $d = 16$, number of labelers $k = 5$, and offline samples $n = 20, 50, 100, 200$. We compare the following approaches: (a) Naive MLEs that simply computes the MLEs given the preference data and optimizes a policy against the average reward estimate to maximize social welfare; (b) Pessimistic Social Welfare [37], which is the pessimistic version of Naive MLEs; (c) Median of MLEs, which optimizes a policy against the reward function derived from the median over MLEs; (d) Pessimistic MoMLEs, which is our proposed algorithm and outlined in Algorithm 1. We report the policy suboptimality under both truthful and strategic labeling.

**Results.**   Overall, we observe that while the Naive MLEs and its pessimistic version, Pessimistic Social Welfare, perform well when labelers are truthful, the performance degrades substantially under strategic preference labeling. In contrast, Pessimistic Median of MLEs exhibits almost no degradation, consistent with its approximate strategyproofness guarantee. Median of MLEs also shows slightly more robustness, though not to the same degree. We find that with increasing sample size, Pessimistic Median of MLEs primarily suffers the inherent cost of being strategyproof, as indicated by diminishing performance gains from more samples. That said, the influence of strategic manipulation on the learned policy also grows with more data, thereby making discouraging strategic preference labeling increasingly more valuable.

Table 1: Suboptimality $\mathrm{SubOpt}(\hat{\pi})$ under truthful and strategic labeling across dataset sizes $n$.

| n = 20 | | | |
|---|---|---|---|
| **Algorithm** | **SubOpt (Truthful)** | **SubOpt (Strategic)** | **Difference** |
| Naive MLEs | $\mathbf{0.512} \pm 0.067$ | $\mathbf{0.649} \pm 0.082$ | $+0.137$ |
| Pessimistic SW | $0.641 \pm 0.081$ | $0.751 \pm 0.086$ | $+0.110$ |
| Median of MLEs | $0.635 \pm 0.053$ | $0.693 \pm 0.167$ | $+0.058$ |
| Pessimistic MoMLEs | $0.661 \pm 0.098$ | $0.715 \pm 0.155$ | $+\mathbf{0.054}$ |

| n = 50 | | | |
|---|---|---|---|
| **Algorithm** | **SubOpt (Truthful)** | **SubOpt (Strategic)** | **Diff.** |
| Naive MLEs | $\mathbf{0.384} \pm 0.056$ | $0.622 \pm 0.142$ | $+0.238$ |
| Pessimistic SW | $0.403 \pm 0.064$ | $0.652 \pm 0.149$ | $+0.249$ |
| Median of MLEs | $0.516 \pm 0.062$ | $0.706 \pm 0.124$ | $+0.190$ |
| Pessimistic MoMLEs | $0.508 \pm 0.056$ | $\mathbf{0.532} \pm 0.154$ | $+\mathbf{0.024}$ |

| n = 100 | | | |
|---|---|---|---|
| **Algorithm** | **SubOpt (Truthful)** | **SubOpt (Strategic)** | **Diff.** |
| Naive MLEs | $\mathbf{0.230} \pm 0.071$ | $\mathbf{0.516} \pm 0.131$ | $+0.316$ |
| Pessimistic SW | $0.249 \pm 0.068$ | $0.584 \pm 0.147$ | $+0.336$ |
| Median of MLEs | $0.522 \pm 0.044$ | $0.605 \pm 0.049$ | $+0.083$ |
| Pessimistic MoMLEs | $0.506 \pm 0.045$ | $0.574 \pm 0.152$ | $+\mathbf{0.068}$ |

| n = 200 | | | |
|---|---|---|---|
| **Algorithm** | **SubOpt (Truthful)** | **SubOpt (Strategic)** | **Diff.** |
| Naive MLEs | $\mathbf{0.133} \pm 0.029$ | $0.491 \pm 0.251$ | $+0.358$ |
| Pessimistic SW | $0.136 \pm 0.027$ | $0.459 \pm 0.266$ | $+0.323$ |
| Median of MLEs | $0.487 \pm 0.061$ | $0.514 \pm 0.259$ | $+0.027$ |
| Pessimistic MoMLEs | $0.474 \pm 0.051$ | $\mathbf{0.415} \pm 0.079$ | $-\mathbf{0.059}$ |

