# OpenReview forum: "Strategyproof Reinforcement Learning from Human Feedback"
_NeurIPS.cc/2025/Conference — NeurIPS 2025 poster_

### Official Review · Reviewer_zDgj · 2025-06-27

**Clarity:** 4
**Significance:** 3
**Originality:** 4
**Rating:** 5
**Confidence:** 3

**Summary:**

This paper tackles the problem of what happens in RLHF when there are multiple labelers who might strategically lie to push the resulting AI's policy closer to their own preferred outcomes. The authors frame this through the lens of mechanism design, looking at how to create systems that are robust to this kind of manipulation.

They begin by formally demonstrating that current RLHF methods aren’t immune to this issue. In fact, they prove that even a single strategic person can cause a surprising amount of distortion to the policy’s alignment with the overall group’s welfare. The most striking part is their impossibility result, where they tie the issue back to classic ideas from social choice theory, arguing that any perfectly strategyproof RLHF system is fundamentally limited in optimality. They therefore points to a core tradeoff between robustness to manipulation and getting the best possible policy. Given this challenge, they propose an alternative algorithm, which idea is to use pessimistic estimates combined with a median rule, which they show is approximately strategyproof and, under the right conditions, still finds the optimal policy as you get more data and more labelers

**Questions:**

* I really enjoyed the theoretical deep dive. To help bridge this theory to practice, I'd be curious to hear your thoughts on the value of a small, synthetic empirical study. How do you think such experiments might help us better understand the algorithm's performance in (a) more "average-case" strategic settings, and (b) the finite-sample regime? I feel like this could really strengthen the paper's message.

* I'd like to push a bit on Assumption 2 (the hyperrectangle spanning assumption). Could you give me a bit more intuition on when you think this might hold, and what you suspect might happen to your guarantees if it's violated?

**Ethical Concerns:**

["NO or VERY MINOR ethics concerns only"]

**Final Justification:**

The synthetic experiments addressed my concerns

**Limitations:**

yes

**Paper Formatting Concerns:**

no concern

**Quality:**

3

**Strengths And Weaknesses:**

# Strengths

* Tackling strategyproofness in RLHF without relying on monetary incentives is, in my opinion, a critical research direction. The discovery of the fundamental trade-off between incentive alignment and policy alignment feels like a really important and lasting conceptual contribution.

* From a technical standpoint, the work appears to be very solid (though I didn’t verify the proofs)

* I also have to commend the authors on the writing: the paper is exceptionally clear and was a pleasure to read. It does a great job of motivating the problem and laying out the contributions in a logical, easy-to-follow manner.

* The proposed "Pessimistic Median of MLEs" algorithm is a novel contribution. Combining pessimistic estimates with a median rule to handle strategic behavior is a clever idea, and the paper provides a solid theoretical analysis for why this approach works.

# Weaknesses

While the theory is elegant, I did find myself wondering about a few practical aspects, which could probably be clarified with some quick synthetic experiments.

* The theoretical guarantees are asymptotic, but as always, the devil’s in the constants. Some of these bounds could hide large constants that render the algorithm impractical for realistic sample sizes. Is it effective with thousands of samples? Or do you need millions? Without some empirical calibration, it’s hard to tell where the method becomes usable.

* Similarly, the proofs focus on specific worst-case constructions. It would be valuable to see how these methods hold up in more "normal" average-case scenarios, where e.g. the $\theta$ are not adversarially chosen, but sampled from some realistic distribution. This would give us a better sense of how brittle current methods are in the wild

---

> ### Author Rebuttal · Authors · 2025-07-30
>
> Thank you very much for the time and effort you spent reviewing our work and your helpful comments. We are especially appreciative of your comments on the importance of strategyproof RLHF without monetary incentives, the conceptual clarity of the tradeoff between incentive and policy alignment, and the novelty of the proposed algorithm.
>
> Below we respond to your comments and questions, where we first jointly respond to your questions and suggestions about adding an experimental evaluation.
>
>
> ### **Adding an Experimental Evaluation:**
> While the paper’s **main contributions are theoretical**, we agree that experiments can help interpret and better understand the guarantees and the dynamics of strategic manipulation in RLHF. We therefore **added synthetic experiments** to the paper, an excerpt of which can be found further below. Following your suggestions and questions, we specifically analyzed the following two aspects:
> 1. **“Dependence on the number of samples?”**
> We provide results for sample sizes $n=20, 50, 100, 200$ with $d=16$ below. In the low sample regime $n=20$, the pessimistic approaches (Pessimistic Median of MLEs and Pessimistic Social Welfare) achieve only about 60% of the optimal policy’s social welfare (1.6 is the optimal social welfare in the experiments), but quickly improve performance with increasing sample size. We find that eventually the main performance limitation of Pessimistic Median of MLEs becomes not the sample size, but instead the inherent cost of being strategyproof (by using a median rule), as indicated by diminishing performance gains from more samples. That said, the influence of strategic manipulation also grows with more data, making strategyproofness increasingly valuable. By $n = 200$, Pessimistic Median of MLEs outperforms all baselines under strategic behavior.
> 2. **“Performance in average-case scenarios (not worst-case)?”**
> You are correct in that the proofs and guarantees are worst-case over possible problem instances. In the newly added experiments, we sample the true reward parameters $\theta_i^\star$ independently from a multivariate Gaussian distribution. From the experiments we observe that even in such average-case scenarios, labelers can have significant incentives to manipulate (see Difference column).
>
>
>
> **“More intuition about Assumption 2; What if it’s violated?”**
> Assumption 2 models the set of achievable feature expectations as a hyperrectangle rather than, for example, a sphere or general convex set. This structure simplifies the analysis and our intuition is that this should act as a reasonable approximation when features are bounded and vary independently across dimensions. In practice, the set $\{ \mathbb{E}_{s \sim \rho}[\phi(s, \pi(s))] : \pi \in \Pi \}$ often occupies a bounded convex subset of $\mathbb{R}^d$, and the hyperrectangle assumption can be seen as an idealized version of such sets. As noted in Footnote 4, our results extend naturally to mild violations of this assumption, with only small additive error terms in the guarantees.
>
>
> Please let us know if any of your concerns or questions remain.
>
>
> ### **Excerpt from Newly Added Experiments**
> **Experimental Setup:**
> We simulate strategic labeling behavior by performing approximate gradient ascent (using SPSA) on each labeler's utility $J_i(\hat \pi)$ w.r.t. the labelers' internal reward parameters $\tilde \theta_i$, which govern their preference distributions $P_{\tilde \theta_i}$. Each labeler is initialized at their ground-truth reward vector $\theta_i^*$, sampled from a multivariate Gaussian. Labeler strategies are optimized for 200 steps. Since this process requires repeatedly relabeling data and rerunning each algorithm, we focus on small settings. All results are averaged over 5 random seeds, and we report standard errors. The results below are for embedding dimension $d = 16$, numer of labelers $k = 5$, and offline samples $n = 20, 50, 100, 200$, and additional runs will be included in the appendix of the paper. We report the suboptimality under both truthful and strategic labeling, along with the difference.
>
> Overall, we observe that while the Naive MLEs and its pessimistic version, Pessimistic Social Welfare, perform well when labelers are truthful, the performance degrades substantially under strategic labeler behavior. In contrast, Pessimistic Median of MLEs exhibits almost no degradation, consistent with its approximate strategyproofness. Median of MLEs also shows slighty more robustness, though not to the same degree.
>
>
> #### n = 20:
>
> | Algorithm                    | $\text{SubOpt}(\hat{\pi})$ (Truthful Labelers) | $\phantom{00} \text{SubOpt}(\hat{\pi})$ (Strategic Labelers) |  $\phantom{00}$ Difference |
> |-----------------------------|----------------------------:|-----------------------------:|----------------------------:|
> | Naive MLEs                  | $0.512 \pm 0.067$                                              | $0.649 \pm 0.082$                                               | $+0.137$                   |
> | Pessimistic Social Welfare  | $0.641 \pm 0.081$                                              | $0.751 \pm 0.086$                                               | $+0.110$                   |
> | Median of MLEs              | $0.635 \pm  0.053$                                              | $0.693 \pm 0.167$                                               | $+0.058$                   |
> | Pessimistic Median of MLEs | $0.661 \pm 0.098$                                              | $0.715 \pm 0.155$                                               | $+0.054$                   |
> |
>
> #### $n=50$
> | Algorithm                    | $\text{SubOpt}(\hat{\pi})$ (Truthful Labelers) | $\phantom{00} \text{SubOpt}(\hat{\pi})$ (Strategic Labelers) |  $\phantom{00}$ Difference |
> |-----------------------------|---------------------------------------------------------------:|----------------------------------------------------------------:|----------------------------:|
> | Naive MLEs                  | $0.384 \pm 0.056$                                              | $0.622 \pm 0.142$                                               | $+0.238$                   |
> | Pessimistic Social Welfare  | $0.403 \pm 0.064$                                              | $0.652 \pm 0.149$                                               | $+0.249$                   |
> | Median of MLEs              | $0.516 \pm 0.062$                                              | $0.706 \pm 0.124$                                               | $+0.190$                   |
> | Pessimistic Median of MLEs | $0.508 \pm 0.056$                                              | $0.532 \pm 0.154$                                               | $+0.024$                   |
> |
>
>
> #### n=100
>
>
> | Algorithm                    | $\text{SubOpt}(\hat{\pi})$ (Truthful Labelers) | $\phantom{00} \text{SubOpt}(\hat{\pi})$ (Strategic Labelers) |  $\phantom{00}$ Difference |
> |-----------------------------|---------------------------------------------------------------:|----------------------------------------------------------------:|----------------------------:|
> | Naive MLEs                  | $0.230 \pm 0.071$                                              | $0.516 \pm 0.131$                                               | $+0.316$                   |
> | Pessimistic Social Welfare  | $0.249 \pm 0.068$                                              | $0.584 \pm 0.147$                                               | $+0.336$                   |
> | Median of MLEs              | $0.522 \pm 0.044$                                              | $0.605 \pm 0.049$                                               | $+0.083$                   |
> | Pessimistic Median of MLEs | $0.506 \pm 0.045$                                              | $0.574 \pm 0.152$                                               | $+0.068$                   |
> |
>
>
> #### n = 200:
>
> | Algorithm                    | $\text{SubOpt}(\hat{\pi})$ (Truthful Labelers) | $\phantom{00} \text{SubOpt}(\hat{\pi})$ (Strategic Labelers) |  $\phantom{00}$ Difference |
> |-----------------------------|---------------------------------------------------------------:|----------------------------------------------------------------:|----------------------------:|
> | Naive MLEs                  | $0.133 \pm 0.029$                                              | $0.491 \pm 0.251$                                               | $+0.358$                   |
> | Pessimistic Social Welfare  | $0.136 \pm 0.027$                                              | $0.459 \pm 0.266$                                               | $+0.323$                   |
> | Median of MLEs              | $0.487 \pm 0.061$                                              | $0.514 \pm 0.259$                                               | $+0.027$                   |
> | Pessimistic Median of MLEs | $0.474 \pm 0.051$                                              | $0.415 \pm 0.079$                                               | $-0.059$                   |
> |

---

> > ### Comment · Reviewer_zDgj · 2025-08-06
> >
> > I thank the author for this convincing rebuttal which addressed my concerns. I will raise the score accordingly.

---

### Official Review · Reviewer_rwWs · 2025-06-30

**Clarity:** 3
**Significance:** 3
**Originality:** 3
**Rating:** 5
**Confidence:** 3

**Summary:**

This paper studies Reinforcement Learning from Human Feedback (RLHF) in settings where labelers may strategically misreport their preferences.

On the negative side, it shows that:
(i) several popular pluralistic-alignment algorithms are manipulable (i.e., not strategy-proof), and
(ii) any algorithm that is fully strategy-proof must, in the worst case, achieve only a 1/k fraction of the optimal social welfare, where k is the number of labelers.

On the positive side, the paper proposes Pessimistic Median of MLEs (MoMLE), a robust offline algorithm that, under linear rewards and certain coverage assumptions, is approximately strategy-proof and nearly optimal when the sample size n is large relative to the feature dimension d and the number of labelers k.

The analysis is first given for contextual bandits, then extended to finite-horizon MDPs.

**Questions:**

I am on the fence between "borderline accept"  and "accept". It would be helpful to clarify the following question to better see how compelling the results are.

1. In the performance guarantee theorems, how do the constants kappa_i and nu_i depend on parameters such as n and d?

2. In Theorem 4.1, the result conditions on the other labelers’ reports, and labeler i’s utility is evaluated ex-ante (i.e., taking expectation over their choices). Plus, the proposed algorithm is deterministic, and J_i is a deterministic function. Given these setups, over what source of randomness is the “with probability at least 1 - delta” statement defined?

3. Related to W1, if a strategic labeler is allowed to deviate from the BT model altogether (e.g., always deterministically selecting the higher-reward response), how would this affect the performance of the proposed algorithm?

4. Conversely, how difficult is it in practice to achieve epsilon-strategyproofness? For example, is the plain median aggregation method (without pessimistic adjustment) already o(1)-strategyproof and nearly optimal?

5. Are Pessimistic Social Welfare and MaxMin-RLHF epsilon strategyproof? Proposition 3.3 only shows that they are not (strictly) strategyproof.

**Ethical Concerns:**

["NO or VERY MINOR ethics concerns only"]

**Final Justification:**

I find the topic to be timely and the solution is clean. Overall, I learned something from the paper and can see a good possibility that it spurs follow-up research on incentive-aware AI alignment. I understand the work is main theoretical but that is in line with the style many papers of similar topics.

My main justification for the score raise is that, based on the author rebuttal, the deviation strategy space can be broader than originally stated in the paper. In addition, the paper does incorporate incentive structures in the policy design and analysis, so it is not just about "delicate ignorance of (or insensitivity to) data".

**Limitations:**

yes

**Paper Formatting Concerns:**

NIL

**Quality:**

3

**Strengths And Weaknesses:**

Strength:

S1. (Timely and promising topic) This paper considers a timely topic. The consideration of strategic manipulation in human feedback to LLM alignment is natural, and I can see a good possibility that it spurs follow-up research on incentive-aware AI alignment.


S2. (Natural design) The proposed algorithm is also natural. Median as a statistic is known for being robust to outliers. It is hence natural that even if an agent can unilaterally change his/her own perceived theta_i, the effect on the aggregated theta will only be minimal, leading to the small incentive to manipulate.


Weakness:

W1. (Solution concept)  The solution concept feels somewhat counterintuitive. On the one hand, the requirement of epsilon-strategyproofness is quite strong, as it demands that truth-telling be an ex-ante dominant strategy rather than simply a Nash equilibrium. On the other hand, the space of deviations is rather restricted: if a labeler deviates, they must still produce responses according to a BT model, albeit with a different parameter theta_i. This excludes arguably benign forms of preference reporting, such as simply choosing the preferred response deterministically (i.e., always selecting the higher-reward option).


W2. (Empirical validation) This paper is purely theoretical and could benefit from the empirical validation. That said, I do not see this as a fundamental flaw.

---

> ### Author Rebuttal · Authors · 2025-07-30
>
> Thank you very much for taking the time to review our paper and your helpful suggestions. We are pleased to hear that you found our research topic timely and promising (S1) and our algorithm design natural (S2). We respond to your questions below.
>
> **W1. (Space of deviations is restricted) & Question 3 (What if labelers deviate from the BT model if they wanted to?):**
> In fact, our approximate strategyproofness result for the Pessimistic Median of MLEs does not rely on the assumption that deviating labelers must follow a BT model. The analysis **extends directly to arbitrary deviation strategies**, that is, labelers can follow any preference distribution if they wanted to. To briefly outline why, in the proof of Theorem 4.1, we show that for any fixed data $D_{-i}$ from the other labelers, the optimal outcome for labeler $i$ in terms of expected utility would be to directly report their true reward parameter $\theta_i^\star$. Since labelers only influence the mechanism through the data they provide, we then show that providing labels according to a BT model with parameter $\theta_i^\star$ yields expected utility within $O(\sqrt{d/n})$ of this best case. This argument does *not* rely on any assumptions about the structure of potential deviations. As for the specific case where a labeler deterministically selects the higher-reward option, this behavior corresponds to a BT model with high-magnitude parameters (rewards) and is therefore already included within our analysis. Thank you very much for highlighting this in your review, as this will be helpful to clarify in the paper. We have added a remark in the main text to make this generalization explicit.
>
>
>
> **W2. (Empirical validation; the paper could benefit from an empirical evaluation):**
> As you already pointed out, our main contributions are **theoretical** and **conceptual**. Still, we appreciate your suggestion, and **added synthetic experiments** to the paper (you can find an excerpt of the results with brief explanations further below).
>
>
> ### Responses to your other Questions:
> 1. **Coverage constants $\kappa_i$ and $\nu_i$:**
> The constants $\kappa_i$ and $\nu_i$ importantly do not depend on the other parameters. This means that there are no other parameter dependencies hidden in these constants.
> 2. **Source of randomness in Theorem 4.1:**
> Thank you for catching this. The source of randomness is meant to come from sampling from $\smash{P_{\tilde \theta_i}}$ and $\smash{P_{\theta_i^\star}}$ and we shouldn’t have written the expectation over these distributions but instead that the labels in $\smash{\tilde D_i}$ are sampled from $P_{\tilde \theta_i}$ and the labels in $D_i^\star$ are sampled from $P_{\theta_i^\star}$. We fixed this.
> 3. **"What if the labelers are allowed to deviate from the BT model?"**
> (Please see the response to W1 above.)
> 4. **“How difficult is it in practice to achieve $\varepsilon$-strategyproofness? What about non-pessimistic median MLE?”**
> This is an interesting question, which we don’t have a fully definitive answer to. What we do know is that in terms of social welfare (i.e., policy suboptimality) the median approach without pessimism will be theoretically **much worse** since pessimism is generally needed for provable sample efficiency: Theorem 1.2 in [35] tells us that the non-pessimistic “naive” MLE approach to RLHF suffers from **constant suboptimality**. Regarding the strategyproofness, our intuition is that the basic median approach should be approximately strategyproof as well (partially confirmed by the newly added experiments). However, a theoretical analysis is quite challenging as it becomes fully probabilistic due to the lack of pessimistic confidence bounds.
> 5. **“Are Pessimistic Social Welfare and MaxMin-RLHF epsilon-strategyproof?”**
> No. We can adapt the examples in the proof of Proposition 3.3 so that for any value of $\varepsilon$ the incentive to manipulate exceeds $\varepsilon$. Note that the current counterexample in the proof already implies  $\varepsilon > 1/4$. We included a short sentence mentioning this below the proposition.
>
> Please let us know if any of your concerns or questions remain. We are happy to further discuss.
>
>
>
> ### **Excerpt from Newly Added Experiments**
> **Experimental Setup:**
> We simulate strategic labeling behavior by performing approximate gradient ascent (using SPSA) on each labeler's utility $J_i(\hat \pi)$ w.r.t. the labelers' internal reward parameters $\tilde \theta_i$, which govern their preference distributions $P_{\tilde \theta_i}$. Each labeler is initialized at their ground-truth reward vector $\theta_i^*$, sampled from a multivariate Gaussian. Strategies are optimized for 200 steps. Since this process requires repeatedly relabeling data and rerunning each algorithm, we focus on small settings. All results are averaged over 5 random seeds, and we report standard errors. The results below are for embedding dimension $d = 16$, numer of labelers $k = 5$, and offline samples $n = 50$. We report the suboptimality under both truthful and strategic labeling, along with the difference.
>
> We observe that while the Naive MLEs and its pessimistic version, Pessimistic Social Welfare, perform well when labelers are truthful, the performance degrades substantially under strategic labeler behavior. In contrast, Pessimistic Median of MLEs exhibits almost no degradation, consistent with its approximate strategyproofness. Median of MLEs also shows slighty more robustness, though not to the same degree.
>
> | Algorithm                    | $\text{SubOpt}(\hat{\pi})$ (Truthful Labelers) | $\phantom{00} \text{SubOpt}(\hat{\pi})$ (Strategic Labelers) |  $\phantom{00}$ Difference |
> |-----------------------------|---------------------------------------------------------------:|----------------------------------------------------------------:|----------------------------:|
> | Naive MLEs                  | $0.384 \pm 0.056$                                              | $0.622 \pm 0.142$                                               | $+0.238$                   |
> | Pessimistic Social Welfare  | $0.403 \pm 0.064$                                              | $0.652 \pm 0.149$                                               | $+0.249$                   |
> | Median of MLEs              | $0.516 \pm 0.062$                                              | $0.706 \pm 0.124$                                               | $+0.190$                   |
> | Pessimistic Median of MLEs | $0.508 \pm 0.056$                                              | $0.532 \pm 0.154$                                               | $+0.024$                   |
> |
>
>
>
> [35] Zhu et al. Principled reinforcement learning with human feedback from pairwise or k-wise comparisons.

---

> > ### Comment · Reviewer_rwWs · 2025-08-03
> > **Connection to Differential Privacy**
> >
> > I thank the authors for their detailed reply and the new numerical experiments are commendable. Most of my earlier questions have been addressed satisfactorily.
> >
> > I have only one follow-up comment regarding the previous W1/Q3. If the analysis and performance guarantees indeed extend to arbitrary deviation strategies, then the "structured deviation" under BL distribution may not be essential. Rather, the key intuition appears to be that the proposed data aggregation algorithm is insensitive to any single labeler's data, which in turn reduces the incentive for any individual to deviate unilaterally. It would be helpful if the authors could confirm whether this high-level intuition is accurate (or clarify what aspects might be missing). If the interpretation holds, the core idea seems conceptually related to the literature on differential privacy, which also seeks to limit the influence of any single participant on the outcome. So some discussion could be helpful.

---

> > > ### Author Response · Authors · 2025-08-03
> > >
> > > > Rather, the key intuition appears to be that the proposed data aggregation algorithm is insensitive to any single labeler's data, which in turn reduces the incentive for any individual to deviate unilaterally. It would be helpful if the authors could confirm whether this high-level intuition is accurate (or clarify what aspects might be missing). If the interpretation holds, the core idea seems conceptually related to the literature on differential privacy, which also seeks to limit the influence of any single participant on the outcome. So some discussion could be helpful.
> > >
> > > Thank you for the follow-up.
> > >
> > > Your high-level intuition is close, but it indeed misses a crucial aspect. Under Pessimistic Median of MLEs, a single labeler’s data can influence the outcome, sometimes substantially. However, and this is the **key point**: no labeler can move the final policy closer to their individually preferred outcome by deviating unilaterally. In other words, they can move the aggregate (in the worst case by a lot), but simply not in their personal favor.
> > >
> > > This property may not be immediately intuitive for complex preferences, policies, and returns in the context of RLHF, but it becomes clearer in simpler settings like resource allocation. Suppose three agents $A$, $B$, and $C$ each report a point on the real line, with true values $a < b < c$, and each agent would like the aggregated point to be as close to their true value as possible. It is quick to check (e.g., by drawing the points and potential misreports on a line) that the median is strategyproof in this setting even though an agent may have substantial influence over the aggregate: any agent who misreports only moves the median further from their own true value.
> > >
> > >
> > > Regarding your earlier intuition, approximate mechanism design and differential privacy are indeed closely related, precisely because your intuition can be put into practical use: if individual influence is limited (as in DP), the incentive to manipulate is also small. There are several great papers on this, e.g., McSherry and Talwar (2007) and Nissim et al. (2012). However, our approach does not fall into this category. Pessimistic Median of MLEs is not differentially private and does not rely on limiting influence in the same way. It is also worth noting that differential privacy typically yields approximate strategyproofness. In our setting, if labelers could report their reward parameters directly, the pessimistic median approach would be exactly strategyproof (unlike DP mechanisms). We obtain approximate strategyproofness because labelers can only report preference labels / data. Even then, if $n \to \infty$, we get exact strategyproofness. Nonetheless, exploring whether DP-style techniques could lead to scalable, approximately strategyproof RLHF is an interesting direction for future work, and we will clarify this distinction in the paper.
> > >
> > > We're happy to discuss further if anything remains unclear.
> > >
> > > F. McSherry and K. Talwar (2007). Mechanism Design via Differential Privacy
> > >
> > > Nissim et al. (2012). Approximately optimal mechanism design via differential privacy

---

> > > > ### Comment · Reviewer_rwWs · 2025-08-04
> > > >
> > > > I thank the authors for the further clarification and discussion. I found the responses to be clear and my questions have been addressed satisfactorily. I plan to raise my score to "accept". Congrats on the nice work.

---

### Official Review · Reviewer_y9WG · 2025-07-02

**Clarity:** 3
**Significance:** 3
**Originality:** 3
**Rating:** 4
**Confidence:** 3

**Summary:**

The paper addresses the vulnerability of RLHF to strategic manipulation by human labelers. The core problem is that when labelers know their feedback can influence the final learned policy, they may misreport preferences to skew the outcome toward their own interests. This is especially problematic when labelers have diverse and potentially conflicting preferences. The key contributions include a formalization of strategic RLHF and definition of strategyproofness. Some negative results are presented, like existing RLHF methods are not strategyproof, and a single strategic labeler can cause the learned policy to deviate arbitrarily far from the optimal social welfare. It also proves that any strategyproof RLHF algorithm must, in the worst case, perform k-times worse than the optimal policy.

**Questions:**

Overall I believe it is an interesting and solid paper. However, I do have the following questions:

1. In terms of motivation, my main concern is whether strategic manipulation in terms of preferences by human labelers is practical. Because in reality, there is often a large number of labelers for training an LLM and as a result, the influence of each labeler's decision on the result is subtle and perhaps imperceptible. It is better to include some concrete motivating example in the paper to show that such strategic manipulations are happening in reality. Otherwise I'm not fully convinced that any human labeler is capable or initiative enough to engage in such manipulations.

2. The paper assumes that all labelers’ preferences are linearly realizable in known feature spaces. This assumption is mathematically convenient but might be overly restrictive in practice. Human preferences are rarely linear or easily projected onto known features. I'm wondering if similar insight or result can hold under different assumptions, e.g., discrete or convex preference structures?

3. The proposed Pessimistic Median of MLEs algorithm uses a conservative pessimistic estimate over the coordinate-wise median reward parameter. While it can promote robustness, it may also lead to overly cautious policies. I'm curious if there is any discussion about the potential tradeoff in such an algorithm design.

**Ethical Concerns:**

["NO or VERY MINOR ethics concerns only"]

**Limitations:**

Yes

**Paper Formatting Concerns:**

No major concern.

Ln. 246, invalid reference

**Quality:**

3

**Strengths And Weaknesses:**

The strengths of the paper:

Interesting and timely research topic, solid theoretical results. The proposed solution combines median aggregation and pessimistic estimates over confidence sets of reward parameters is interesting.

The weaknesses of the paper:

Motivation not strong enough, involves restrictive assumptions, lack experimental support.

---

> ### Author Rebuttal · Authors · 2025-07-30
>
> Thank you for taking the time to review our work. We are happy to hear that you found the research topic interesting and timely with solid theoretical results, and our proposed method interesting.
>
> Below, we address your concerns and questions about the **motivation** (what about large numbers of labelers), **assuming linearly realizable rewards**, and **experimental support** (we now added synthetic experiments):
>
> 1. **Motivation ("What if the number of labelers is very large?"):**
> When the number of individual labelers in RLHF is large, it is often more appropriate to think in terms of **groups of labelers** whose preferences closely align, for example along political, ideological, or stylistic lines. Each such group can be treated as a single effective labeler in our framework. Notably, such group effects have been observed in practice, where ChatGPT models became more politically biased after RLHF fine-tuning, which suggests that the human labelers’ personal preferences influenced the model [19, 29]. If a group is aware of this potential impact, it may have an incentive to strategize to steer the model toward its values. Moreover, single labelers (small groups) can have **substantial influence** even in large labeler populations. Proposition 3.4 shows this theoretically, and recent work by Baumgärtner et al. (2024) demonstrates empirically that injecting as little as 1-5% manipulated data is sufficient to shift a model's behavior toward targeted sentiment generation. Additional evidence from Betley et al. (2025) shows that fine-tuning on a narrow task with a strategic or malicious annotator can cause general misalignment. We now highlight these points more clearly in the introduction to strengthen the motivation.
> 2. **Assuming Linearly Realizable Rewards:**
> We agree that assuming linearly realizable rewards can be a restrictive assumption that may be violated in practice. For more complex reward models, our ideas could likely be extended, as MLE concentration results exist for more general models (Zhan et al. 2023). The pessimistic median based algorithm might not be strategyproof but we believe some variant of the algorithm might work for general reward models and this would be interesting to explore in the future.
> 3. **Pessimistic Estimates and Trade-Offs:**
> We agree with your intuition that pessimism may lead to conservative policy choices, however, it is necessary to some extent in offline RL(HF) to obtain sample efficient algorithms: Theorem 1.2 in [35] shows that **without pessimism** the “naive” MLE approach suffers from **constant suboptimality** in offline RLHF. We don’t have a definitive answer to potential trade-offs, but our **newly added experiments** compare also with non-pessimistic versions of the median and social welfare algorithms (see further below).
>
>
> We would be happy to further discuss if any concerns or questions remain.
>
> ### **Excerpt from Additional Experiments**
> **Experimental Setup:**
> We simulate strategic labeling behavior by performing approximate gradient ascent (using SPSA) on each labeler's utility $J_i(\hat \pi)$ w.r.t. the labelers' internal reward parameters $\tilde \theta_i$, which govern their preference distributions $P_{\tilde \theta_i}$. Each labeler is initialized at their ground-truth reward vector $\theta_i^*$, sampled from a multivariate Gaussian. Strategies are optimized for 200 steps. Since this process requires repeatedly relabeling data and rerunning each algorithm, we focus on small settings. All results are averaged over 5 random seeds, and we report standard errors. The results below are for embedding dimension $d = 16$,  numer of labelers $k = 5$, and offline samples $n = 50$. We report the suboptimality under both truthful and strategic labeling, along with the difference.
>
> We observe that while the Naive MLEs and its pessimistic version, Pessimistic Social Welfare, perform well when labelers are truthful, the performance degrades substantially under strategic labeler behavior. In contrast, Pessimistic Median of MLEs exhibits almost no degradation, consistent with its approximate strategyproofness. Median of MLEs also shows slighty more robustness, though not to the same degree.
>
> | Algorithm                    | $\text{SubOpt}(\hat{\pi})$ (Truthful Labelers) | $\phantom{00} \text{SubOpt}(\hat{\pi})$ (Strategic Labelers) |  $\phantom{00}$ Difference |
> |-----------------------------|---------------------------------------------------------------:|----------------------------------------------------------------:|----------------------------:|
> | Naive MLEs                  | $0.384 \pm 0.056$                                              | $0.622 \pm 0.142$                                               | $+0.238$                   |
> | Pessimistic Social Welfare  | $0.403 \pm 0.064$                                              | $0.652 \pm 0.149$                                               | $+0.249$                   |
> | Median of MLEs              | $0.516 \pm 0.062$                                              | $0.706 \pm 0.124$                                               | $+0.190$                   |
> | Pessimistic Median of MLEs | $0.508 \pm 0.056$                                              | $0.532 \pm 0.154$                                               | $+0.024$                   |
> |
>
>
> [35] Zhu et al., 2023. Principled reinforcement learning with human feedback from pairwise or k-wise comparisons.
>
> Zhan et al., 2023. Provable Offline Preference-Based Reinforcement Learning
>
> Baumgärtner et al. 2024, Best-of-Venom: Attacking RLHF by Injecting Poisoned Preference Data.
>
> Betley et al., 2025. Emergent Misalignment: Narrow fine-tuning can produce broadly misaligned LLMs

---

> > ### Comment · Reviewer_y9WG · 2025-08-05
> > **Official Comment by Reviewer**
> >
> > I thank the authors for the response, they all make sense to me. I decide to maintain my score for weak acceptance.

---

### Official Review · Reviewer_395W · 2025-07-03

**Clarity:** 3
**Significance:** 2
**Originality:** 2
**Rating:** 4
**Confidence:** 3

**Summary:**

The paper is concerned with strategic manipulation in RLHF. It represents a model for the problem. It then shows that some existing RLHF are not strategy proof. In fact, there is a trade-off between strategy proofness and optimality. A strategy proof algorithm (Pessimistic median of MLEs) is then shown to be approximately strategy proof and to have a bounded sub-optimality.

**Questions:**

Please see the points under weaknesses.

**Ethical Concerns:**

["NO or VERY MINOR ethics concerns only"]

**Final Justification:**

I have not changed my recommendation score. The rebuttal has not changed my view of the paper significantly. While it has a nice theoretical contributions, the practical and experimental contributions can be improved significantly especially given that the paper is motivated by such practical applications.

**Limitations:**

Sufficient discussion was included.

**Paper Formatting Concerns:**

I did not notice any formatting issues

**Quality:**

3

**Strengths And Weaknesses:**

## Strengths

-The paper is well-written. The sections follow a good logical sequence.

-I'm not familiar with the area of strategic manipulations in RLHF but this paper seems to give some good results that are missing.


## Weaknesses

-Although, there has been literature on strategic manipulation in RLHF, but a broader point is that it's not clear (at least to me) how much of a practical issue it even is?

-The paper is highly theoretical and no experiments are given. While the results are interesting, it's not clear how realistic they are. Here are more detailed points:

* The dimension d is an embedding dimension for the state-action, correct? What value should we expect for it? Is it larger than the number of samples n we expect to have? This effects many of the given bounds

* How realistic is assumption 2? Is it not reasonable to expect that the set partially covers the space instead of spanning R^d?

* Lines (306-307): “Overall, as the number of samples increases and as the number of labelers grows, the Pessimistic Median of MLEs algorithm converges to the optimal policy”. Looking at proposition 4.3, it’s not clear to me that the upper bound converges to 0? I would expect that even with very large n there would be a lower bound ratio between k and n since any labeler can only label a finite number of points, so the second term in the upper bound would still be on the order of \sqrt{d}, correct?

* Lines (88-89) ”we propose a strategy-robust RLHF method that does not rely on payments or other financial incentives, which are often impractical in real-world applications”

The paper states that using financial incentives is impractical, but in standard RLHF aren’t the labelers paid for the labels they do? So, aren’t financial incentives already involved?

* There is a restriction in the definition of  P_{\tilde{\theta_i}}, I can see a labeler following a distribution other than the truthful one that does not simply imply changing the vector \theta and sampling according to Eq (1).

---

> ### Author Rebuttal · Authors · 2025-07-30
>
> Thank you for taking the time and effort to review our work and your helpful comments. We are glad to read that you found our results interesting and to be a good addition to the literature. We respond to your concerns and questions below.
>
>
> 1. **“Is strategic manipulation of RLHF a practical issue?”**
> We believe that strategic manipulation is a serious concern in practice, which is becoming increasingly important. We want to highlight previous work [29, 19] that showed ChatGPT models exhibiting more politically biased behavior after RLHF fine-tuning, which suggests that the human evaluators’ personal preferences influenced the model. One major take-away from this is that such real-world biases in who provides feedback have had measurable effects on model behavior. Other recent work also shows that even small-scale strategic manipulation of human feedback can have disproportionately large effects on LLM fine-tuning in actual RLHF pipelines [Baumgärtner et al. 2024, Haider et al. 2025], which our Proposition 3.4 also supports theoretically. There is also other recent evidence that fine-tuning on a narrow task with a strategic / malicious annotator can cause general misalignment  [Betley et al. 2025].
>
>
> 2. **“The paper is highly theoretical and no experiments are given.”**
> While the paper’s main contributions are theoretical, we now **added synthetic experiments** to the paper to corroborate our theoretical results, following your and the other reviewers' suggestions. You can find selected results of the newly added experiments further below.
> We now also respond to your additional questions concerning **"How realistic are the results?"**:
>    1. **“Typical values for the dimension d?”**
> Yes, $d$ is the embedding dimension of the state-action pairs and its value really depends on the application. If these embeddings are derived from a neural network or LLM the dimension can be quite large, e.g., 512 to 4096 for LLMs. Compared to this, most open-source preference datasets include anywhere between 5-50k comparisons. We agree with your intuition that our bounds should be interpreted in relation to $d$ and $n$ and if $d \gg n$ offline RL(HF) bounds can become vacuous. However, this is expected and the $O(\sqrt{d/n})$ bound in offline RLHF is tight as discussed in Remark 4.4 and Corollary 4.5.
>    2. **”How realistic is Assumption 2? Is it reasonable to expect that the set partially covers the space instead of spanning $\mathbb{R}^d$?”**
> Assumption 2 models the set of achievable feature expectations as a hyperrectangle rather than, for example, a sphere or general convex set. This structure simplifies the analysis and should act as a reasonable approximation when features are bounded and vary independently across dimensions. In practice, the set $ \{ \mathbb{E}_{s \sim \rho}[\phi(s, \pi(s))] : \pi \in \Pi \}$ often occupies a bounded convex subset of $\mathbb{R}^d$, and the hyperrectangle assumption can be seen as an idealized version of such sets. As noted in Footnote 4, our results extend naturally to mild violations of this assumption, with only small additive error terms in the guarantees.
>    3. **“It's not clear [to me] that the upper bound of Proposition 4.3 converges to 0.”**
> In the mentioned paragraph we are referring to the asymptotic behavior of the algorithm and the corresponding bounds, which means we are looking at the case where the labelers can label an *arbitrarily* large amount of points even though of course this amount will always remain finite in practice. We wanted to express with this statement that if we could keep asking an increasing amount of labelers and each labeler for an arbitrarily large amount of data points, then the algorithm converges. We agree that the paragraph can be confusing and we rephrased the sentence in the paper to clarify this.
>    4. **“Aren’t financial incentives already involved? Why incentive design without payments?”**
> While labelers are typically paid per annotation, implementing financial mechanisms that ensure strategyproofness in RLHF is often impractical. For instance, existing methods [24, 32] require conditioning payments or penalties on downstream effects, such as the performance of a fine-tuned policy, which are only observed well after labeling is complete. This makes enforcement difficult in real-world RLHF pipelines. Moreover, some RLHF settings, such as community-driven or crowdsourced alignment, do not involve financial compensation at all. In these cases, strategyproofness without payments is necessary by design. We also note that achieving strategyproofness without payments is significantly harder than when utility can be transferred directly through financial incentives.
>    5. **“Can labelers deviate to preference distributions other than the BT model?”**
> While we define $P_{\tilde \theta_i}$ via the BT model of Eq. (1), our approximate strategyproofness results in fact do not rely on this specific parameterization. In the proof of Theorem 4.1, we show that for any fixed data from other labelers, the best possible outcome for labeler $i$ would be to directly report their true reward vector ${\theta_i^\star}$. Since labelers can only influence the mechanism through data, we argue that reporting according to $P_{\theta_i^\star}$ is nearly as good, within $O(\sqrt{d/n})$, regardless of what other preference distributions are allowed. This argument extends to arbitrary preference distributions, not just the BT model. We have added a remark in the main text to clarify this generality. Thank you for drawing our attention to this point.
>
> Please let us know if any concerns or questions remain.
>
> ### **Example from Newly Added Experiments**
> **Experimental Setup:**
> We simulate strategic labeling behavior by performing approximate gradient ascent (using SPSA) on each labeler's utility $J_i(\hat \pi)$ w.r.t. the labelers' internal reward parameters $\tilde \theta_i$, which govern their preference distributions $P_{\tilde \theta_i}$. Each labeler is initialized at their ground-truth reward vector $\theta_i^*$, sampled from a multivariate Gaussian. Strategies are optimized for 200 steps. Since this process requires repeatedly relabeling data and rerunning each algorithm, we focus on small settings. All results are averaged over 5 random seeds, and we report standard errors. The results below are for embedding dimension $d = 16$, numer of labelers $k = 5$, and offline samples $n = 50$. We report the suboptimality under both truthful and strategic labeling, along with the difference.
>
> We observe that while the Naive MLEs and its pessimistic version, Pessimistic Social Welfare, perform well when labelers are truthful, the performance degrades substantially under strategic labeler behavior. In contrast, Pessimistic Median of MLEs exhibits almost no degradation, consistent with its approximate strategyproofness. Median of MLEs also shows slighty more robustness, though not to the same degree.
>
> | Algorithm                    | $\text{SubOpt}(\hat{\pi})$ (Truthful Labelers) | $\phantom{00} \text{SubOpt}(\hat{\pi})$ (Strategic Labelers) |  $\phantom{00}$ Difference |
> |-----------------------------|---------------------------------------------------------------:|----------------------------------------------------------------:|----------------------------:|
> | Naive MLEs                  | $0.384 \pm 0.056$                                              | $0.622 \pm 0.142$                                               | $+0.238$                   |
> | Pessimistic Social Welfare  | $0.403 \pm 0.064$                                              | $0.652 \pm 0.149$                                               | $+0.249$                   |
> | Median of MLEs              | $0.516 \pm 0.062$                                              | $0.706 \pm 0.124$                                               | $+0.190$                   |
> | Pessimistic Median of MLEs | $0.508 \pm 0.056$                                              | $0.532 \pm 0.154$                                               | $+0.024$                   |
> |
>
>
>
> [19] Hartmann et al., 2023. The political ideology of conversational AI.
>
> [24] Park et al., 2024. RLHF from heterogeneous feedback via personalization and preference aggregation.
>
> [29] Santurkar et al., 2023. Whose opinions do language models reflect?
>
> [32] Sun et al., 2024. Mechanism design for LLM fine-tuning with multiple reward models.
>
> Baumgärtner et al., 2024, Best-of-Venom: Attacking RLHF by Injecting Poisoned Preference Data.
>
> Haider et al., 2025. A framework for mitigating malicious RLHF feedback in LLM training using consensus based reward.
>
> Betley et al., 2025. Emergent Misalignment: Narrow fine-tuning can produce broadly misaligned LLMs

---

### Decision · Program_Chairs · 2025-09-17

**Decision:**

Accept (poster)

**Comment:**

The authors study strategyproofness of RLHF. In their model, labelers may strategically misreport feedback to steer the policy toward their own preferences. They show that existing RLHF algorithms are not strategyproof and that, in the worst case, any fully strategyproof RLHF algorithm must perform k times worse than the optimal policy, where k is the number of labelers. However, by slightly relaxing strategyproofness, they can do better: in line with previous work in social choice on the strategyproofness of median rules, they propose the Pessimistic Median of MLEs algorithm, which is approximately strategyproof and converges to the optimal policy in the limit. The central observation is while any single labeler can still move the policy an arbitrary amount, they can never move it closer to their preferences by reporting untruthfully.

Strengths:
- All reviewers appreciated the strong theoretical results in the paper.
- The paper is well-written overall.
- To address reviewer concerns (395W, y9WG, rwWs, zDgj) about empirical evaluation, the authors ran experiments and presented results in the rebuttal. All of the reviewers appreciated them, and the experiments support the theory in the paper.

Weaknesses:
- There was some disagreement about how practically relevant this problem is. How often are labelers actively trying to steer RLHF algorithms toward their own opinion? In their rebuttal, the authors provided a fairly convincing argument about this happening in political topics.
- The paper focuses on worst-case constructions as opposed to average-case scenarios; this was partially addressed by synthetic experiments.